# Lmx1b is required at multiple stages to build expansive serotonergic axon architectures

Lauren J Donovan, William C Spencer, Meagan M Kitt, Brent A Eastman, Katherine J Lobur, Kexin Jiao, Jerry Silver, Evan S Deneris*

Department of Neurosciences, School of Medicine, Case Western Reserve University, Cleveland, United States

**Abstract** Formation of long-range axons occurs over multiple stages of morphological maturation. However, the intrinsic transcriptional mechanisms that temporally control different stages of axon projection development are unknown. Here, we addressed this question by studying the formation of mouse serotonin (5-HT) axons, the exemplar of long-range profusely arborized axon architectures. We report that LIM homeodomain factor 1b (Lmx1b)-deficient 5-HT neurons fail to generate axonal projections to the forebrain and spinal cord. Stage-specific targeting demonstrates that Lmx1b is required at successive stages to control 5-HT axon primary outgrowth, selective routing, and terminal arborization. We show a Lmx1b→Pet1 regulatory cascade is temporally required for 5-HT arborization and upregulation of the 5-HT axon arborization gene, Protocadherin-alphac2, during postnatal development of forebrain 5-HT axons. Our findings identify a temporal regulatory mechanism in which a single continuously expressed transcription factor functions at successive stages to orchestrate the progressive development of long-range axon architectures enabling expansive neuromodulation.

DOI: https://doi.org/10.7554/eLife.48788.001

*For correspondence:
esd@case.edu

Competing interests: The authors declare that no competing interests exist.

## Introduction

Newly generated neurons dramatically transform their morphology to establish mature circuit connectivity. Some neurons, for example interneurons, connect to local circuits and thus need to extend their axons relatively short distances. In contrast, other neuron types, such as those giving rise to neuromodulatory systems, have the capacity to extend extremely long axons to innervate distant target fields. This is well exemplified in the case of serotonin (5-HT) synthesizing neurons. 5-HT modulates the excitability of nearly all neural circuitry in the brain and spinal cord despite being produced in a relatively small population of neurons (*Azevedo et al., 2009*; *Baker et al., 1991*; *Hornung, 2003*). Expansive serotonergic neuromodulation is achieved through the formation of long-range highly diffuse ascending and descending projection pathways that deliver 5-HT throughout the brain and spinal cord for synaptic or non-synaptic interactions with an array of receptors (*Hannon and Hoyer, 2008*; *Steinbusch, 1981*). Despite the decades-long knowledge of 5-HT's importance as a global neuromodulator, it is not understood how these small numbers of neurons develop, over an extended period of morphological maturation, such elaborate axonal architectures.

5-HT neurons initiate axon outgrowth concomitant with their birth and onset of 5-HT synthesis (*Hawthorne et al., 2010*; *Lidov and Molliver, 1982*). Classic neuroanatomical studies have defined three successive stages, collectively spanning several weeks, in the formation of 5-HT projection pathways: primary pathway formation during which 5-HT axon outgrowth is initiated, selective pathway routing, and finally postnatal terminal arborization (*Lidov and Molliver, 1982*). Following primary pathway outgrowth through the medial forebrain bundle (MFB), ascending 5-HT axons

originating in the dorsal raphe (DRN), median raphe (MRN) and B9 groups of midbrain/pons 5-HT neurons are selectively routed along pre-existing fiber tracts and reach all forebrain targets at parturition in an unarborized state (*Bang et al., 2012*; *Lidov and Molliver, 1982*; *Muzerelle et al., 2016*). Descending 5-HT axons, originating in the medullary serotonergic clusters, raphe pallidus (RPa), raphe obscurus (ROb), and raphe magnus (RMg), enter the spinal cord via the dorsolateral and ventral funiculi. These primary projections then route medially to invade nearly all lamina of the dorsal and ventral horns as well as the intermediate zone from cervical to sacral levels (*Rajaofetra et al., 1989*). After the major ascending and descending projection pathways are formed, a final, entirely postnatal, stage of 5-HT projection pathway maturation ensues during which 5-HT axons originating in different anatomically defined sub-regions flourish profuse terminal arbors in complementary and topographically organized patterns (*Muzerelle et al., 2016*; *Ren et al., 2018*). Terminal arborization develops at least through the first four postnatal weeks and leaves few, if any, regions of the brain and spinal cord devoid of serotonergic input (*Gagnon and Parent, 2014*; *Hornung, 2003*; *Lidov and Molliver, 1982*; *Maddaloni et al., 2017*; *Steinbusch, 1981*). What are the intrinsic mechanisms that govern successive stages in the formation of long-range profusely arborized 5-HT axon projection pathways and how are they temporally coordinated? One possibility is that each stage is governed by different intrinsic regulatory factors. Alternatively, a single continuously expressed intrinsic regulator may act at successive stages to orchestrate progressive morphological maturation of 5-HT pathways.

In contrast to the poor understanding of how 5-HT axonal pathways are formed, there is substantial knowledge of the gene regulatory networks (GRNs) that generate 5-HT neurons and control acquisition of 5-HT transmitter identity (*Deneris and Gaspar, 2018*). The LIM homeodomain protein, Lmx1b, is a crucial factor in 5-HT GRNs as 5-HT neuron selective targeting of *Lmx1b* results in the failure to induce Tph2 expression for 5-HT synthesis and Slc6a4 expression for 5-HT reuptake (*Zhao et al., 2006*). This results in extremely low levels of 5-HT in the adult brain, which is associated with high neonatal mortality and several abnormal behavioral phenotypes including hyperactivity, delayed respiratory maturation, enhanced inflammatory pain sensitivity, deficient opioid analgesia, sleep regulation, and increased contextual fear memories (*Dai et al., 2008*; *Hodges et al., 2009*; *Zhang et al., 2018*; *Zhao et al., 2007a*; *Zhao et al., 2007b*).

Lmx1b is a continuously expressed, terminal selector-type factor in 5-HT neurons (*Hobert, 2008*) raising the possibility that subsequent to its initial role in the induction of 5-HT synthesis and transport it may perform additional stage specific functions in the maturation of serotonergic connectivity. However, stage specific functions of continuously expressed terminal selectors, such as Lmx1b, in postmitotic neuronal morphological maturation are poorly understood (*Deneris and Hobert, 2014*; *Hobert, 2016*). Here, we report that lack of Lmx1b results in the failure to build long-range ascending and descending 5-HT axon projection pathways. Using temporal conditional targeting approaches we dissect distinct stage-specific functions for Lmx1b. Our findings show that Lmx1b acts at successive stages to control primary pathway growth rate, selective pathway routing and terminal arborization of 5-HT axons. We identify an ascending-specific Lmx1b-controlled regulatory cascade that regulates selective pathway routing and then switches to control forebrain 5-HT axon arborization through stage specific expression of genes required for arborization. This study demonstrates that a single continuously expressed transcription factor, initially required for induction of 5-HT synthesis and reuptake, subsequently acts at successive stages to build the expansive axon pathway architectures enabling CNS-wide serotonergic neuromodulation.

## Results

### Lmx1b controls formation of ascending 5-HT projection pathways

Conditional targeting of Lmx1b with the *Pet1-Cre* transgene results in loss of endogenous 5-HT neuron markers, Sert, Tph2, and 5-HT at E12.5 (*Zhao et al., 2006*). Therefore, we generated control (*Lmx1b*[+/+;Pet1-Cre;Ai9]) and *Lmx1b*cKO (*Lmx1b*[fl/fl;Pet1-Cre;Ai9]) mice in which the Ai9 reporter allele was used as a surrogate marker to specifically label Pet1[+] cell bodies and their axons with red fluorescent protein, TdTomato. In control adult mice, 82% of Tph2[+] neurons in the DRN and MRN express TdTomato. Conversely, approximately 88% of TdTomato[+] cells co-label with Tph2; the remaining

TdTomato$^+$ cells likely express Tph2, but at a level insufficient for detection with standard IHC (*Deneris and Gaspar, 2018*; *Okaty et al., 2015*) (*Figure 1—figure supplement 1A,B*).

TdTomato$^+$ cells were not found outside of the raphe nuclei (*Figure 1—figure supplement 1B*). TdTomato$^+$ axons in control mice were widely distributed throughout the adult brain and colocalized with 5-HT (*Figure 1—figure supplement 1C,D*). The pattern of TdTomato$^+$ axons in the control forebrain corresponded closely with the pattern of 5-HT axon distribution previously determined in the rat with an anti-5-HT antibody (*Lidov and Molliver, 1982*; *Steinbusch, 1981*). Similarly, TdTomato$^+$ axon distribution in control brains was highly concordant with 5-HT axon projection patterns determined more recently using a mouse line in which GFP was knocked into the Tph2 coding region, thus validating our surrogate marking of 5-HT axons using a soluble reporter (*Migliarini et al., 2013*).

Abundant numbers of TdTomato$^+$ cell bodies were detected in each of the raphe nuclei of *Lmx1b*cKO animals (*Figure 1—figure supplement 1E*). The vast majority of *Lmx1b*cKO TdTomato$^+$ cells did not co-express Lmx1b or Tph2 (*Figure 1—figure supplement 1E,F*). Further, RT-qPCR analyses verified severe deficits of *Lmx1b* and *Tph2* mRNAs in flow sorted YFP-labeled Pet1$^+$ neurons in *Lmx1b*cKO mice, confirming appropriate Lmx1b targeting (*Figure 1—figure supplement 1G*). Counts of TdTomato$^+$ cell bodies in *Lmx1b*cKO mice indicated that equivalent numbers were present compared to controls at 3 months and that their numbers remained stable for at least 13 months (*Figure 1—figure supplement 1H,I*). *Lmx1b*cKO TdTomato$^+$ cell bodies were located within the normal cytoarchitectural boundaries of the raphe nuclei with a mildly altered distribution and smaller size (*Figure 1—figure supplement 1I,J*). Together, these data indicate specific fluorescent labeling of 5-HT neurons with Ai9 and efficient Lmx1b knock-down in *Lmx1b*cKO animals.

Analysis of TdTomato$^+$ axons in adult *Lmx1b*cKO mice revealed a dramatically different pattern compared to that in control mice (*Figure 1A,B*). Although TdTomato$^+$ axons were present in normal density and with proper ascending trajectory within the MFB of *Lmx1b*cKO mice, TdTomato$^+$ axons were nearly completely missing throughout the forebrain of *Lmx1b*cKO animals distal to the thalamus (*Figure 1B*). Indeed, the vast majority of TdTomato$^+$ axons failed to reach various distal fiber tracts including the fimbria-fornix, supracallosal stria, cingulum bundle, and olfactory tract (*Figure 1B*). Consequently, few if any TdTomato$^+$ axons were present in the olfactory bulb, cortex, amygdala, hippocampus, striatum or many regions of the hypothalamus (*Figure 1B–D*; *Figure 1—figure supplement 2A,B*). It is likely that the few TdTomato$^+$ 5-HT axons that did reach the distal forebrain were the result of a small number of 5-HT neurons in which Ai9 expression was activated, but Lmx1b targeting failed (*Figure 1—figure supplement 1F*).

Although *Lmx1b*cKO TdTomato$^+$ axons were present in the thalamus, the distribution of their terminal arbors was quite different from that in controls (*Figure 1B,E*). We found aberrant clumping of mutant TdTomato$^+$ terminal arbors in several intralaminar thalamic nuclei (*Figure 1E*; *Figure 1—figure supplement 2C*). Many regions were completely devoid of arbors unlike the relatively homogeneously tiled distribution in the control thalamus (*Figure 1E*, arrowheads). In addition, the conspicuously dense 5-HT arborization that demarcates the paraventricular nucleus of the thalamus (PVT) was completely absent in *Lmx1b*cKO mice (*Figure 1E*; *Figure 1—figure supplement 2D*).

To verify the deficit of TdTomato$^+$ axons in the forebrain of *Lmx1b*cKO mice, we performed a second method of axon labeling by stereotaxic injection of an AAV2 virus expressing a Cre-dependent membrane bound channelrhodopsin with a YFP tag (rAAV2/Ef1a-DIO-hchR2-EYFP) into the midbrain of adult control and *Lmx1b*cKO mice. In *Lmx1b*cKO mice, we found an absence of YFP$^+$ axons in all of the distal areas in which TdTomato$^+$ axons were absent. Moreover, YFP$^+$ axons were colocalized in all forebrain areas with TdTomato$^+$ axons (*Figure 1—figure supplement 1K,L*).

## Lmx1b controls formation of descending 5-HT projection pathways

As Lmx1b is strongly expressed in all medullary raphe 5-HT neurons, we next investigated descending 5-HT axon development. In control animals, 80% of Tph2$^+$ medullary cell bodies expressed TdTomato and 92% of TdTomato$^+$ cell bodies were co-labeled with Tph2 (*Figure 2—figure supplement 1A*). TdTomato$^+$ axon patterns in control E15.5 embryos corresponded closely with the pattern of developing 5-HT-labeled axons in the ventral and lateral funiculi of the spinal cord, thus indicating specific labeling of descending 5-HT axons with TdTomato (*Figure 2—figure supplement 1B*). In addition, the pattern of control TdTomato$^+$ axon distribution throughout the embryonic and adult spinal cord corresponded to descending 5-HT axons patterns in gray and white matter

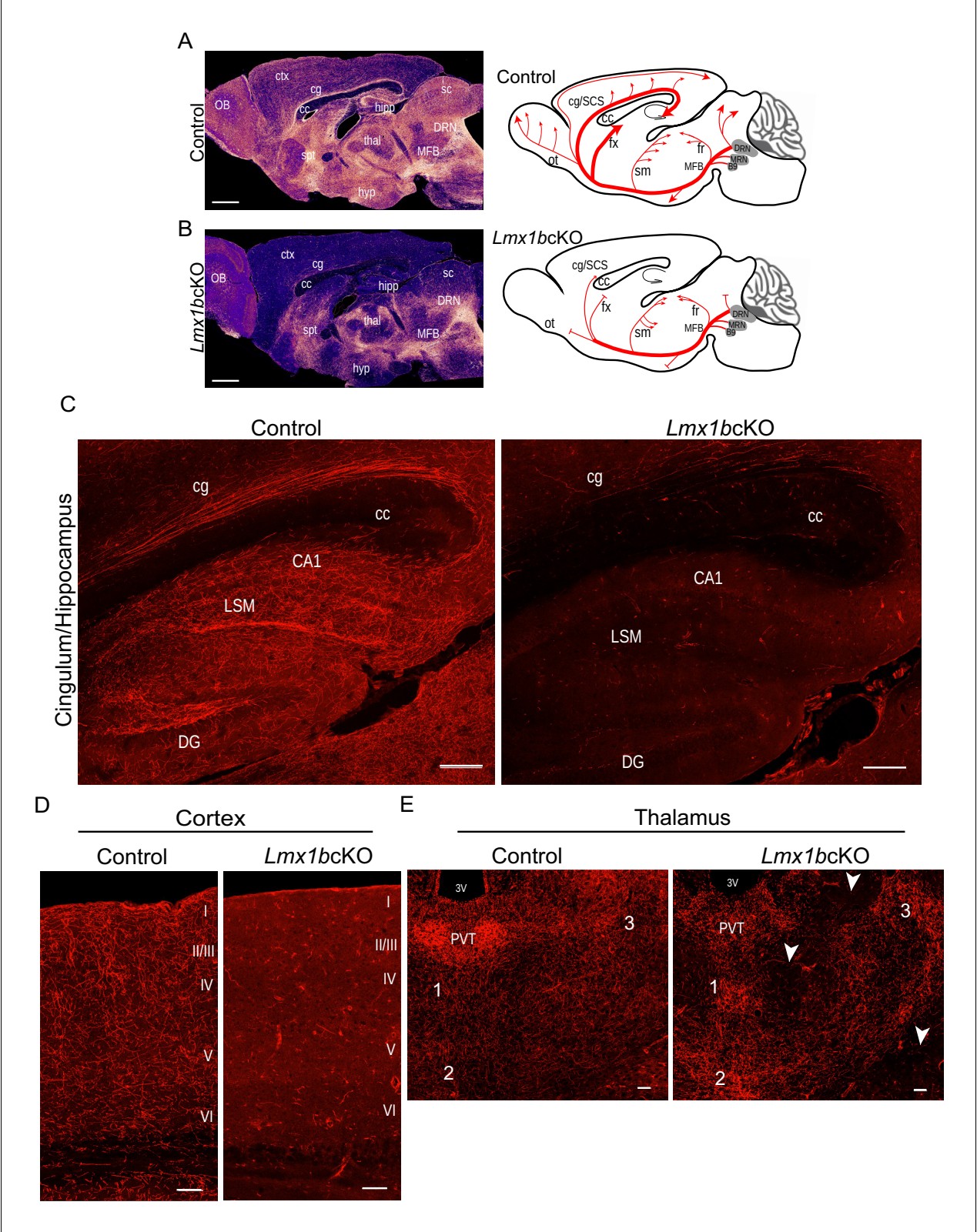

**Figure 1.** Lmx1b is required for the formation of ascending 5-HT axon projection pathways. (A, B) Ascending 5-HT axonal projection system immunolabeled using an anti-RFP antibody to TdTomato in whole sagittal forebrain sections of 3 month old mice displayed by heatmap. *Lmx1b*cKO TdTomato+ axons were nearly absent in numerous brain regions (B) compared to controls (A) (n = 6, controls; n = 7, *Lmx1b*cKO adult mice). *Right*, schematics depicting 5-HT axon trajectories in *Lmx1b*cKO vs. control brains. Scale bars, 1000 μm. OB, olfactory bulb; ctx, *cortex*; cg, *cingulum*; cc,

*Figure 1 continued on next page*

*Figure 1 continued*

*corpus callosum*; hipp, *hippocampus*; spt, *septum*; hyp, *hypothalamus*; thal, *thalamus*; sc, *superior colliculus*; MFB, *medial forebrain bundle*; DRN, *dorsal raphe nucleus*. Schematic (*right*): ot, *olfactory tract*; cg/SCS, *cingulum/supracallosal stria*; fx, *fornix*; sm, *stria medularis*; fr, *fasciculus retroflexus*. (C) Confocal images of TdTomato$^+$ axons in sagittal sections. *Lmx1bcKO* axons failed to fill cingulum bundles or innervate the hippocampus. Scale bars, 200 μm. cg, *cingulum*; cc, corpus callosum; LSM, *lacunosum moleculare*; DG, *dentate gyrus*; CA1 of hippocampus. (D) Coronal sections of cortex show near complete lack of *Lmx1bcKO* TdTomato$^+$ axons Scale bars, 50 μm. (E) Coronal view of altered patterns of TdTomato$^+$ axons in *Lmx1bcKO* thalamus. Arrowheads indicate areas devoid of axons in *Lmx1bcKO* thalamus. Numbers correspond to areas of axon clumping in *Lmx1bcKO* thalamus. See **Figure 1—figure supplement 2** for high magnification images. Scale bars, 100 μm. PVT, *paraventricular nucleus of the thalamus*; 3V, *third ventricle*.

DOI: https://doi.org/10.7554/eLife.48788.002

The following figure supplements are available for figure 1:

**Figure supplement 1.** Surrogate marking of 5-HT cell bodies and axons and Lmx1b conditional targeting.
DOI: https://doi.org/10.7554/eLife.48788.003
**Figure supplement 2.** Lmx1b deficiency disrupts 5-HT axon patterns in the forebrain.
DOI: https://doi.org/10.7554/eLife.48788.004

described in the rat and mouse with an anti-5-HT antibody (*Ballion et al., 2002*; *Rajaofetra et al., 1989*). We confirmed that the number of TdTomato-labeled Pet1$^+$ neurons in the medullary raphe nuclei did not differ between *Lmx1bcKO* and control mice at 3 months of age (*Figure 2—figure supplement 1C*). Further, *Lmx1bcKO* TdTomato$^+$ cells did not express Tph2, confirming targeting of Lmx1b (*Figure 2—figure supplement 1D*).

We found a severe lack of TdTomato$^+$ axons in the white matter funiculi, through which 5-HT axon projections normally extend caudally through the spinal cord (*Figure 2A*; *Figure 2—figure supplement 2A*). In addition, a severe reduction of *Lmx1bcKO* TdTomato$^+$ axons was found in gray matter at all levels of the spinal cord (*Figure 2A*). The normally dense innervation to the dorsal and ventral horns of control spinal cords was dramatically decreased in cervical, thoracic and lumbar levels in *Lmx1bcKO* mice (*Figure 2A*; *Figure 2—figure supplement 2B,C*). At cervical levels there was a 67% deficit of TdTomato$^+$ axons in white matter while at lumbar levels there was a 92% deficit (*Figure 2B*). Quantitation of TdTomato$^+$ axons in gray matter revealed a progressive cervical to lumbar deficit, 73% and 94%, respectively in *Lmx1bcKO* cords compared to control cords (*Figure 2C*). Collectively, our results indicate that Lmx1b deficiency severely disrupts long-range 5-HT axon architecture in the forebrain and spinal cord and that the deficit becomes increasingly profound with increasing distance from 5-HT cell bodies.

## Delayed primary pathway formation and aborted selective pathway routing in *Lmx1bcKO* mice

We next followed the development of *Lmx1bcKO* TdTomato$^+$ axons to distinguish among several possible explanations for the dramatic defects in ascending and descending 5-HT axon pathways: i, disrupted primary pathway formation; ii, failure to selectively route axons through pre-existing tracts; iii, abnormal axon trajectories with subsequent failure to extend, iv, normal pathway development followed by dieback. We first investigated TdTomato$^+$ axons at E13.5 when primary pathway formation is underway and Lmx1b conditional targeting has occurred. At this stage, TdTomato$^+$ axon outgrowth and trajectory in *Lmx1bcKO* mice were normal as they coursed through the MFB over the mesencephalic flexure (*Figure 3A*). At E16.5, however, *Lmx1bcKO* TdTomato$^+$ axons exhibited a clear failure to extend as far rostrally as control axons, suggesting either aborted or delayed axon outgrowth along the more rostral portions of the MFB during late stage primary pathway formation (*Figure 3B*). At E18.5, the density of control TdTomato$^+$ axons was maximal throughout the MFB and TdTomato$^+$ axons were present as far rostral as the septum and diagonal band, where 5-HT axons turn dorsally to navigate through the cingulum, supracallosal stria, and fimbria-fornix (*Figure 3C*). *Lmx1bcKO* TdTomato$^+$ axons now appeared at normal density throughout the MFB, thus supporting delayed axon outgrowth at E16.5 (data not shown). However, at this stage, most *Lmx1bcKO* TdTomato$^+$ axons failed to turn dorsally and remained stalled within the MFB. A smaller number of *Lmx1bcKO* TdTomato$^+$ axons did turn dorsally away from the MFB to pass through the septum and diagonal band (*Figure 3C*). Virtually all *Lmx1bcKO* axons failed to extend and selectively route into the cingulum and fimbria-fornix, thus failing to reach the cortex and hippocampus.

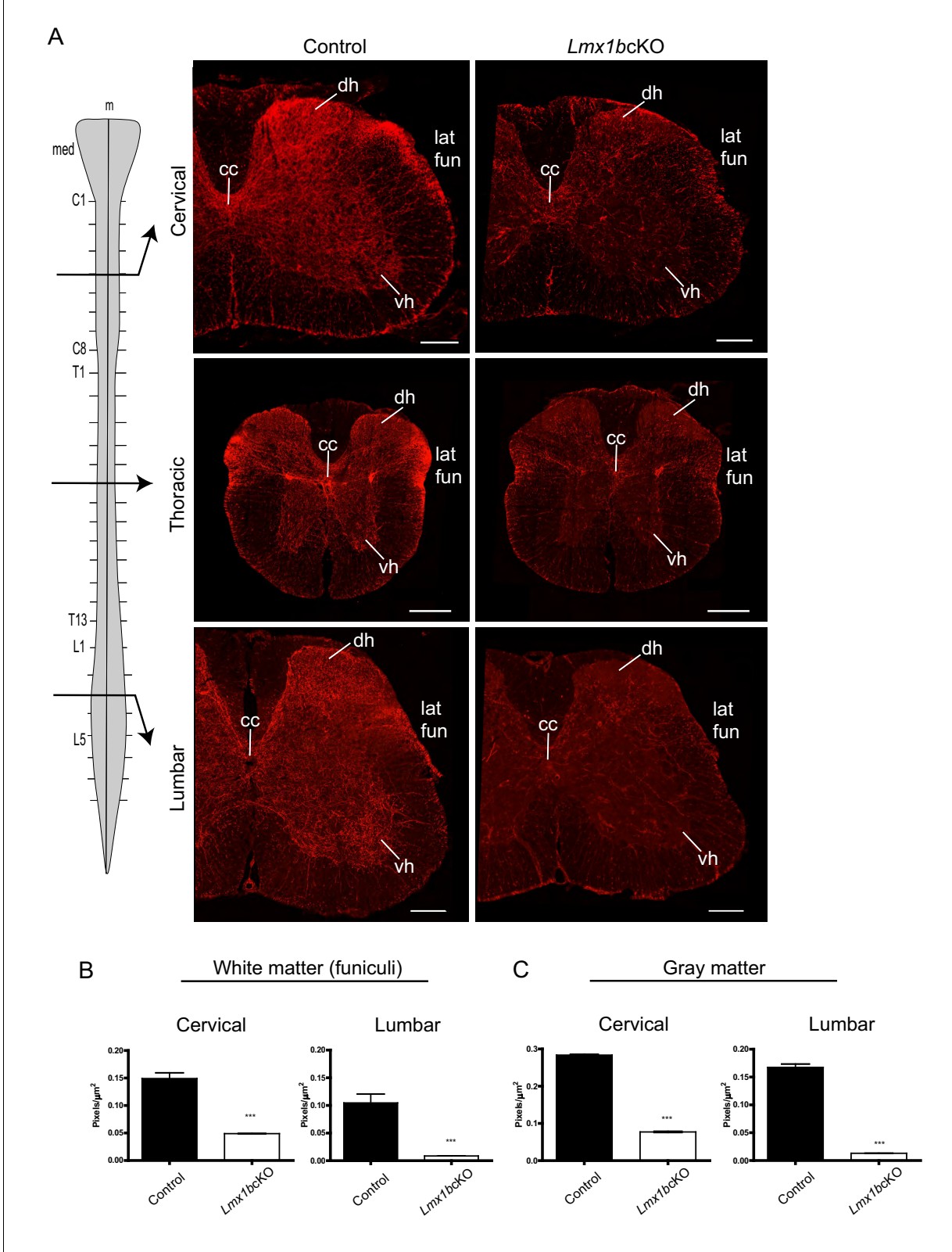

**Figure 2.** Lmx1b is required for the formation of descending 5-HT axon projection pathways.  (**A**) Coronal sections taken at cervical (C4), thoracic (T6), and lumbar (L3) levels of the spinal cord (diagram, *left*). Immunolabeling for TdTomato shows *Lmx1b*cKO axons were severely reduced at every level of the cord in both gray and white matter compared to controls. Scale bars, 200 μm. m, *midline*; med, *medulla*; cc, *central canal*; dh, *dorsal horn*; vh, *ventral horn*; lat fun, *lateral funiculi*. (**B, C**) Quantification of total TdTomato[+] axons (pixels/μm$^2$) in white (**B**) and gray (**C**) matter at cervical and lumbar

*Figure 2 continued on next page*

*Figure 2 continued*

levels (n = 3, control; n = 3 *Lmx1b*cKO mice). Two-way ANOVA with Welch's correction, *p<0.05, **p<0.001, and ***p<0.0001. Data are represented as mean ± SEM.

DOI: https://doi.org/10.7554/eLife.48788.005

The following figure supplements are available for figure 2:

**Figure supplement 1.** Conditional targeting of Lmx1b in the descending 5-HT projection pathway.

DOI: https://doi.org/10.7554/eLife.48788.006

**Figure supplement 2.** Progressive deficits of 5-HT axon fibers in Lmx1b deficient spinal cord white and gray matter.

DOI: https://doi.org/10.7554/eLife.48788.007

*Lmx1b*cKO TdTomato$^+$ axons still did not exhibit ectopic trajectories. Further examination in one-month old *Lmx1b*cKO animals revealed a similar deficit of TdTomato$^+$ axons in distal forebrain tracts (data not shown). These findings suggest that conditional targeting of Lmx1b at E12.5 results in delayed primary pathway formation followed by a profound failure of ascending axons to selectively route into pre-existing fiber tracts.

Analysis of descending TdTomato$^+$ fibers at E15.5 revealed similar initial axon outgrowth through the caudal medulla to the upper cervical spinal cord from *Lmx1b*cKO and control medullary 5-HT cell bodies (*Figure 3D*). Although *Lmx1b*cKO TdTomato$^+$ axons appropriately entered the lateral and ventral funiculi of the cervical spinal cord, greatly reduced densities were evident beginning at mid-cervical levels (*Figure 3D*). At lumbar levels, *Lmx1b*cKO TdTomato$^+$ axons were nearly undetectable in the lateral and ventral funiculi (*Figure 3D*). These results indicate that TdTomato$^+$ axons were not able to extend into the funiculi, which caused a severe and progressive cervical to lumbar deficit of 5-HT innervation in the adult *Lmx1b*cKO spinal cord (*Figures 2* and *3D*).

## Lmx1b acts temporally to control 5-HT axon selective pathways

As formation of 5-HT axon projection pathways occurs over several weeks of embryonic to early postnatal neural maturation we next sought to determine whether Lmx1b is temporally required for 5-HT axon pathway formation. To address this question, we developed a tamoxifen inducible targeting strategy (*Figure 4A*) to knock-down Lmx1b at different early postnatal timepoints. Thus, we generated *Lmx1b*icKO (*Lmx1b*$^{fl/fl}$; *Tph2-CreER*; Ai9) and iControl (*Lmx1b*$^{+/+}$; *Tph2-CreER*; Ai9) mice. Subcutaneous delivery of tamoxifen into iControl pups resulted in approximately 94% of Tph2$^+$ neurons co-labeled with TdTomato in the DRN/MRN/B9 (*Figure 4—figure supplement 1A,B*). Conversely, 94% of TdTomato$^+$ cells were co-labeled with Tph2 (*Figure 4—figure supplement 1B*). As with the *Pet1-Cre* driver, the *Tph2-CreER* activated TdTomato$^+$ cells that are Tph2$^-$ evidently express Tph2 at a low level (*Deneris and Gaspar, 2018*; *Okaty et al., 2015*). RT-qPCR analysis of flow sorted TdTomato$^+$ cells verified that *Lmx1b* mRNA was significantly reduced in *Lmx1b*icKO mice (*Figure 4B*). Equivalent numbers of TdTomato-labeled cell bodies were generated in *Lmx1b*icKO versus iControl mice after injection of tamoxifen (*Figure 4—figure supplement 1C,D*). In contrast to *Lmx1b*cKO TdTomato$^+$ cells, cell body size and distribution of tamoxifen treated *Lmx1b*icKO TdTomato$^+$ cells were not different from that of iControl TdTomato$^+$ cells (*Figure 4—figure supplement 1C,E*).

We analyzed TdTomato$^+$ axons in P1 targeted *Lmx1b*icKO mice four weeks after tamoxifen delivery, when 5-HT axon pathways have fully matured (*Lidov and Molliver, 1982*; *Maddaloni et al., 2017*). We found a dramatic deficit in TdTomato$^+$ arbors throughout the hippocampus in *Lmx1b*icKO mice compared to controls (*Figure 4C*). In addition, TdTomato$^+$ arbors were severely reduced in all layers of the cortex with the most medial layers almost completely devoid of arbors in *Lmx1b*icKO mice (*Figure 4D*). These results clearly demonstrate that Lmx1b is temporally required for postnatal development of 5-HT terminal axons. However, we noticed that in addition to reduced 5-HT arbors in the hippocampus and cortex, TdTomato labeling within the supracallosal stria and cingulum, major routes to the hippocampus and cortex, were consistently less intense in P1 targeted *Lmx1b*icKO mice compared to controls (*Figure 4C*, asterisk). Indeed, further detailed examination revealed that these major 5-HT routes had not yet fully formed (*Figure 4E*). Quantification of TdTomato$^+$ axons within the cingulum and supracallosal stria tracts confirmed a significant decrease in P1 targeted *Lmx1b*icKO mice compared to controls (*Figure 4F*). These results indicate that i,

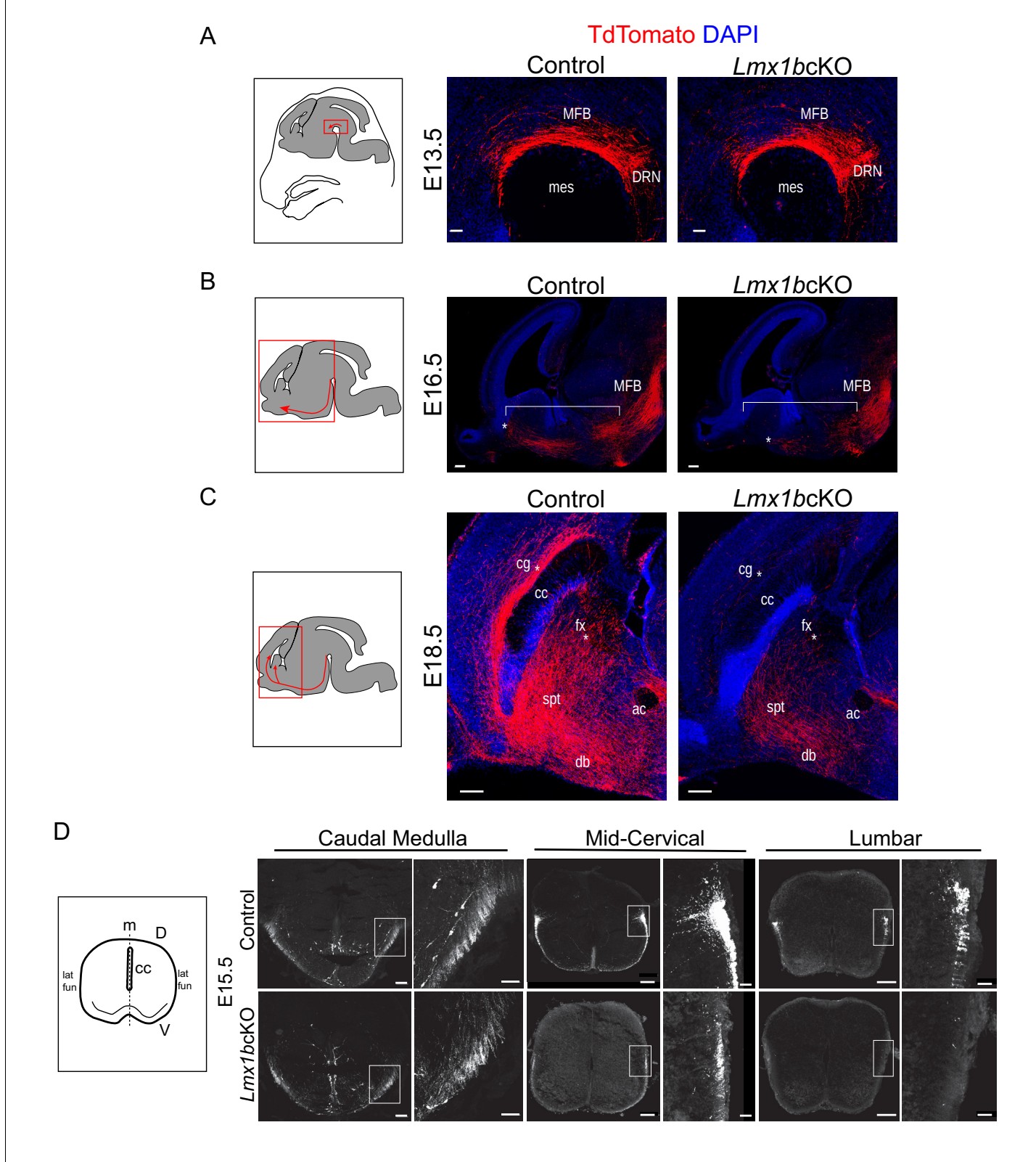

**Figure 3.** Initial axon outgrowth is delayed and selective pathway routing fails in Lmx1b deficient 5-HT neurons. (A–C) Immunolabeled TdTomato+ ascending axons in sagittal slices at different embryonic stages. Diagrams (*left*) show area of image (red box) presented for each time point. Arrows indicate direction of growing axons. E13.5 *Lmx1b*cKO axons exhibited similar ascending trajectories and densities as controls (A). E16.5 *Lmx1b*cKO axons did not extend as far (*asterisk*) and were less abundant (under bracket) compared to control axons (B). E18.5 *Lmx1b*cKO axons failed to fill
*Figure 3 continued on next page*

Figure 3 continued

multiple axon tracts (cg, fx; *asterisks*) compared to controls (**C**). Scale bars, 50 µm (**A**), 200 µm (**B,C**). DRN, *dorsal raphe nucleus*; MFB, *medial forebrain bundle*; mes, *mesencephalic flexure*; cg, *cingulum bundle*; fx, *fornix*; ac, *anterior commissure*; spt, *septum*; db, *diagonal band*; cc, *corpus callosum*. (**D**) Diagram (*left*) depicting coronal section of an embryonic spinal cord. TdTomato$^+$ descending axons at E15.5 in control vs *Lmx1b*cKO embryos. *Lmx1b*cKO axons exit caudal medulla similar to controls but were severely reduced in funiculi at lower levels of the cord (mid-cervical and lumbar). Boxed region of lateral funiculi enlarged to the right of each image. Scale bars, 100 µm (low magnification), 50 µm (high magnification-medulla), 20 µm (high magnification- cervical/lumbar insets). Lat fun, *lateral funiculi*; cc, *central canal*; m, *midline*; D, *dorsal*; V, *ventral*.
DOI: https://doi.org/10.7554/eLife.48788.008

long-range 5-HT axon routing is continuing to develop in the early neonatal period and ii, Lmx1b is required at this stage for completion of 5-HT axon selective routing.

## Lmx1b switches function to control terminal arborization

We next targeted *Lmx1b*icKO mice at P3 to further investigate temporal requirements for Lmx1b (*Figure 5A*). Importantly, in contrast to findings obtained in P1 targeted mice, we found that the density of TdTomato-labeled fibers in the supracallosal stria was now similar in P3 targeted *Lmx1b*icKO and iControl mice (*Figure 5—figure supplement 1A*). Furthermore, TdTomato-labeled fibers in the cingulum, one of the longest of 5-HT forebrain tracts, was now comparable in P3 targeted *Lmx1b*icKO and iControl mice (*Figure 5B*). Quantification of the cingulum and supracallosal stria confirmed no significant difference between P3 targeted *Lmx1b*icKO and iControl mice (*Figure 5—figure supplement 1B*). These findings indicate that 5-HT axon routing was complete in P3 targeted *Lmx1b*icKO mice.

Despite fully developed 5-HT selective pathway routes in the forebrain, TdTomato$^+$ arbors were significantly decreased throughout the hippocampus (*Figure 5C*; *Figure 5—figure supplement 1B*). A majority of remaining arbors detected within the hippocampus of P3 targeted *Lmx1b*icKO mice were found in the molecular layer (SLM), which are the first 5-HT arbors to appear in the hippocampus during rat development (*Lidov and Molliver, 1982*). TdTomato$^+$ arbors were also severely decreased in all cortical layers of the P3 targeted *Lmx1b*icKO brain including in layer I and VI where axons of passage also exist (*Figure 5D*) (*Lidov and Molliver, 1982*). Quantification of the cortex confirmed a significant decrease of TdTomato$^+$ axon arbors in P3 targeted *Lmx1b*icKO compared to iControls (*Figure 5—figure supplement 1B*). It is important to note that many 5-HT terminal arbors have already been generated in the forebrain at P3 (*Lidov and Molliver, 1982*; *Maddaloni et al., 2017*). Thus, Lmx1b targeting at P3 results in the failure of new arbors to form while leaving previously generated arbors intact.

Interestingly, arborization deficits were also found within the thalamus of P3 targeted *Lmx1b*icKO mice. Similar to what we observed in *Lmx1b*cKO mice, the normally dense arborization within the PVT was absent in P3 targeted *Lmx1b*icKO mice, rendering the PVT indiscernible (*Figure 5E*). Together, these results reveal a successive stage of continuous Lmx1b function during which it switches to control terminal arborization of forebrain 5-HT axons.

## Targeting of 5-HT synthesis does not impair formation of forebrain and spinal cord 5-HT arbors

Lack of 5-HT itself has been reported to affect development of forebrain 5-HT terminal arbor patterns (*Migliarini et al., 2013*). Therefore, we next investigated whether the arborization defects present in Lmx1b deficient mice resulted from reduced Tph2 expression and consequently reduced 5-HT levels. To specifically knock down Tph2 to a level comparable to that in *Lmx1b*cKO mice, we generated *Tph2*cKO (*Tph2*$^{fl/fl;Pet1-Cre;Ai9}$) and control (*Tph2*$^{+/+;Pet1-Cre;Ai9}$) mice. The numbers of TdTomato$^+$ cells in each raphe nuclei did not differ in *Tph2*cKO and control mice (*Figure 6A*). In the DRN/MRN/B9 nuclei, we found a 74% reduction of Tph2$^+$ neurons in *Tph2*cKO compared to the 79% reduction of Tph2$^+$ neurons in *Lmx1b*cKO mice (*Figure 6B,C*). Additionally, we confirmed loss of 5-HT itself in *Tph2*cKO mice (*Figure 6D*).

We analyzed TdTomato$^+$ terminal arbors throughout the forebrain in adult *Tph2*cKO mice. We did not find differences in *Tph2*cKO versus control TdTomato-labeled arbor patterns anywhere in the forebrain including the hippocampus, cortex, and the PVT (*Figure 6E*). Further, despite loss of Tph2 in medullary neurons of *Tph2*cKO mice (*Figure 6F*), analysis of the spinal cord from cervical to

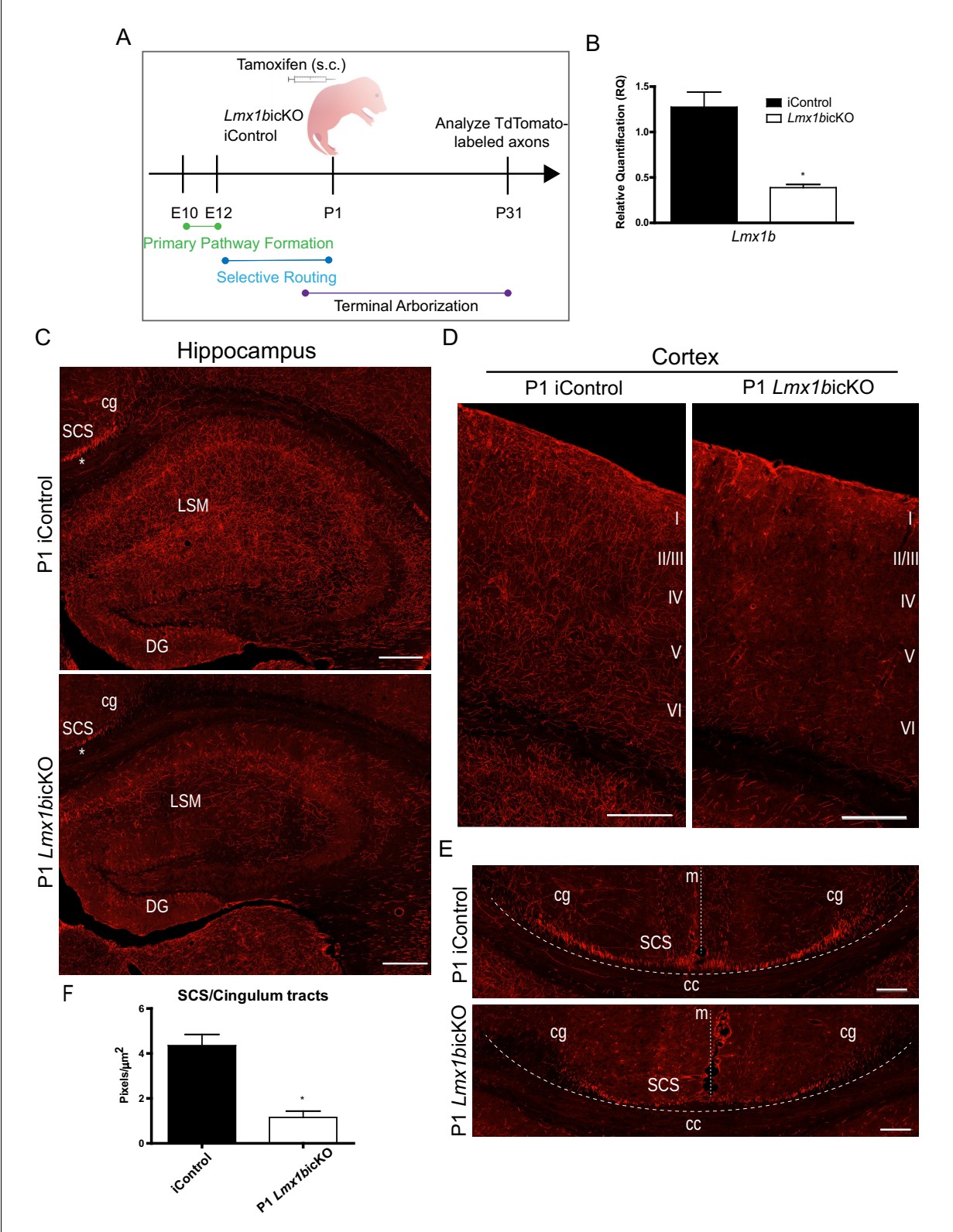

**Figure 4.** Lmx1b is temporally required for 5-HT projection pathway formation. (**A**) Schematic of tamoxifen-inducible approach to target *Lmx1b* at postnatal day (P)1. (**B**) RT-qPCR of flow sorted TdTomato+ neurons from postnatal targeted mice (n = 3, iControl; n = 4, *Lmx1b*icKO mice). Unpaired t-test with Welch's correction, *p<0.05. Data are represented as mean ± SEM. (**C**) Coronal sections of P1 targeted *Lmx1b*icKO hippocampus compared to iControls analyzed at P31. *, incomplete formation of SCS and cingulum in P1 targeted *Lmx1b*icKO brain. Scale bars, 200 µm. (**D**) Coronal sections of

*Figure 4 continued*

P1 targeted *Lmx1b*icKO cortex compared to iControls analyzed at P31. Scale bars, 200 μm. (**E**) Coronal sections at level of corpus callosum showing incomplete formation of major 5-HT axon routes, SCS and cingulum, in P1 targeted *Lmx1b*icKO forebrain compared to iControls (above dotted line). Scale bars, 100 μm. cg, *cingulum*; SCS, *supracallosal stria*; cc, *corpus callosum*; m, *midline*. (**F**) Quantification of axons within SCS and cingulum tracts (n = 3, iControl; n = 3, *Lmx1b*icKO mice). Unpaired t-test with Welch's correction, p=0.0112. Data are represented as mean ± SEM.
DOI: https://doi.org/10.7554/eLife.48788.009

The following figure supplement is available for figure 4:

**Figure supplement 1.** Efficiency of postnatal tamoxifen inducible targeting of Lmx1b.
DOI: https://doi.org/10.7554/eLife.48788.010

lumbar levels revealed no differences in axon densities compared to control mice (*Figure 6G*). Collectively, these results indicate that the loss of Tph2 and thus reduced levels of 5-HT in *Lmx1b*cKO or *Lmx1b*icKO mice did not contribute to either the routing or arborization defects. Evidently, Tph2 and 5-HT levels need to be reduced to a greater extent than we achieved in either our *Lmx1b*cKO or our *Tph2*cKO mice to generate the 5-HT arborization defects observed in mice engineered to completely eliminate Tph2 function and brain 5-HT (*Migliarini et al., 2013*).

## Lmx1b controlled axon-related transcriptomes

Genome-wide analysis of Lmx1b controlled serotonergic transcriptomes has not been performed and consequently only a few serotonergic genes are known Lmx1b targets (*Zhao et al., 2006*). We performed RNA-sequencing (RNA-seq) on flow sorted rostral hindbrain 5-HT neurons, which give rise to the ascending system, and caudal hindbrain 5-HT neurons, which give rise to the descending system, obtained from *Lmx1b*cKO and control E17.5 embryos (*Figure 7A*). Differential expression analysis revealed that the known Lmx1b target genes, *Tph2, Slc18a2, Slc6a4, SCG II, Ctr*, were significantly decreased in our E17.5 rostral and caudal *Lmx1b*cKO datasets. Moreover, we found significantly decreased expression of other 5-HT pathway genes, *Ddc, Slc22a3, Htr1a, Gch1, Gchfr*, in both rostral and caudal *Lmx1b*cKO sorted 5-HT neurons that were previously unknown Lmx1b targets (*Figure 7B,C*).

In addition to these changes, expression of many other genes was altered in the rostral and caudal sorted neurons (*Figure 7D,E*). In rostral *Lmx1b*cKO neurons, expression of 784 genes were significantly decreased, while 118 genes showed significantly increased expression (*Figure 7F,G*). In caudal *Lmx1b*cKO neurons, expression of 1529 genes were significantly decreased and 772 were significantly increased (*Figure 7F,G*). Although there were many genes commonly regulated by Lmx1b, many more were uniquely regulated by Lmx1b in rostral versus caudal 5-HT neurons (*Figure 7F–H*).

Over-representation analysis of Gene Ontology (GO) terms from rostral and caudal Lmx1b-regulated 5-HT neuron genes revealed a pattern of strong enrichment for genes involved in axon development and morphogenesis (*Figure 7I*). In fact, the top enriched GO term, axon development, represents over 150 genes with a Benjamini-Hochberg (BH) adjusted p-value of 0. Many other axon-related terms included axon part, cell leading edge, regulation of supramolecular fiber organization, neuron projection fasciculation, tissue migration, cell-substrate adhesion, negative chemotaxis, morphogenesis of a branching structure, cell leading edge, extracellular matrix binding, and semaphorin receptor binding were enriched at FDR ≤ 0.05. Grouping the genes from all of these axon-related categories together produced a dataset of 422 genes regulated by Lmx1b in rostral and/or caudal 5-HT neurons that have known roles in axon development or function (*Supplementary file 1*). Sixty-six of these downstream genes were significantly altered in both the ascending and descending 5-HT subsystems; 71 genes were uniquely altered in the ascending subsystem while 285 were uniquely altered in the descending subsystem. This analysis suggests that Lmx1b coordinates distinct axonal regulatory programs and transcriptomes to build the two divergent axonal subsystems.

Previous studies have implicated several effector genes in the growth and patterning of 5-HT axons (*Chen et al., 2017*; *Donovan et al., 2002*; *Fournet et al., 2010*; *Katori et al., 2017*; *Lee et al., 2005*). In most cases, however, is it not known whether these genes perform cell intrinsic functions in 5-HT neurons. Examination of our rostral and caudal RNA-seq datasets revealed significantly decreased expression of most but not all of these genes (*Figure 7J,K*). We validated the expression changes of these genes with RT-qPCR from flow-sorted E17.5 rostral 5-HT neurons (*Figure 7O*). Together, these expression studies suggest Lmx1b controls ascending and descending

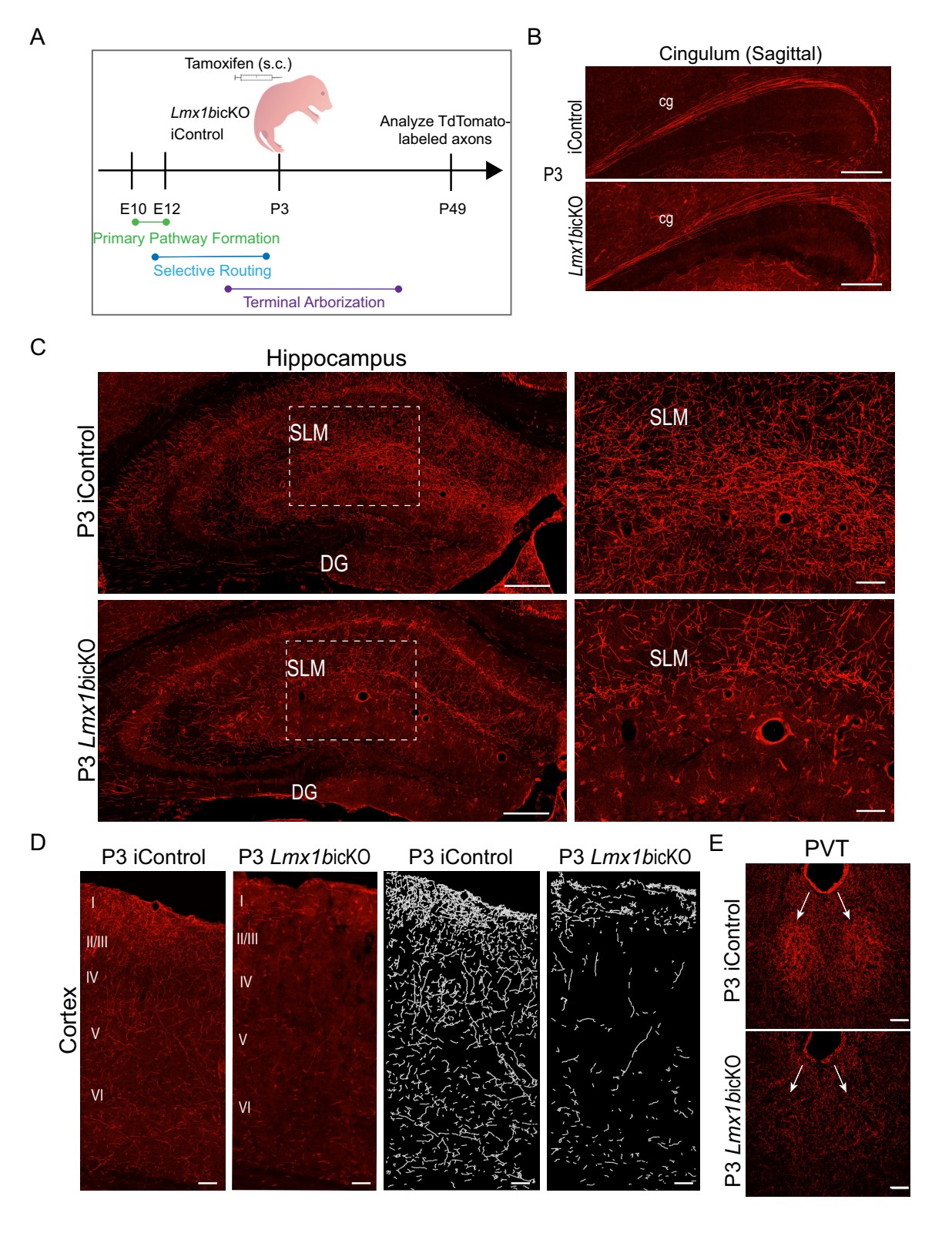

**Figure 5.** Lmx1b temporally controls postnatal 5-HT terminal arborization. (A) Schematic of tamoxifen inducible targeting of *Lmx1b* at postnatal day (P) 3. (B) Sagittal view of cingulum shows fully formed long-range axon routes in P3 targeted *Lmx1b*icKO mice compared to iControls. Scale bars, 200 μm. (C) Coronal sections of hippocampus in P3 targeted *Lmx1b*icKO mice compared to iControls. Dashed boxed region: higher magnification image at right highlighting reduced TdTomato⁺ axons in *Lmx1b*icKO SLM. Scale bars, 200 μm (low magnification), 50 μm (high magnification). SLM, *stratum*

*Figure 5 continued*

*lacunosum moleculare*; DG, *dentate gyrus*. (D) Coronal sections of cortex of P3 targeted *Lmx1b*icKO mice compared to iControls. Imaris tracing; *right panels*. Scale bars, 100 μm. (E) Decreased TdTomato$^+$ arbors detected in P3 targeted *Lmx1b*icKO PVT compared to iControls (arrows). Scale bars, 50 μm.

DOI: https://doi.org/10.7554/eLife.48788.011

The following figure supplement is available for figure 5:

**Figure supplement 1.** P3 targeted *Lmx1b*icKO mice display normal 5-HT axon routing but decreased 5-HT terminal arbors.

DOI: https://doi.org/10.7554/eLife.48788.012

5-HT projection pathways in part through *Gap43, Pcdhac2, Ndn, Ret, Ntrk2, Map6 (STOP), and Celsr3* but that potentially hundreds of other functionally diverse Lmx1b-controlled genes are likely involved in the formation of 5-HT projection pathways (*Figure 7J,K,O*; *Supplementary file 1*).

## An ascending-specific axonal Lmx1b→Pet1 regulatory cascade

The regulatory interactions among transcription factors in the 5-HT GRN are poorly understood but are likely to be important for the development of 5-HT neurons. We reasoned that further study of these interactions might illuminate the regulatory mechanisms through which Lmx1b temporally controls formation of ascending and descending 5-HT axon pathways. Our whole genome expression analyses revealed complex effects of Lmx1b deficiency on the expression of other regulatory factors in the 5-HT GRN (*Figure 7L–O*). Decreased *Pet1* expression was the most consistent and persistent change among the regulatory factors in the rostral and caudal 5-HT GRNs after conditional targeting of Lmx1b, which raised the possibility that an Lmx1b→Pet1 regulatory cascade acts in the formation of 5-HT axons.

To investigate this idea, we generated *Pet1*cKO (*Pet1*$^{fl/fl;Pet1-Cre;Ai9}$) and control (*Pet1*$^{+/+;Pet1-Cre;Ai9}$) mice (*Liu et al., 2010*). We imagined three alternative experimental outcomes: i, *Pet1*cKO fully phenocopies the axon defects found in *Lmx1b*cKO supporting an exclusive Lmx1b→Pet1 regulatory cascade in 5-HT axon formation, ii, 5-HT axon projection pathways are intact in *Pet1*cKO mice and therefore the Lmx1b→Pet1 regulatory path is not operational in 5-HT axon development or iii, *Pet1*cKO incompletely phenocopies *Lmx1b*cKO axon defects suggesting that the Lmx1b→Pet1 cascade operates in parallel with other Lmx1b orchestrated regulatory programs.

We first examined TdTomato$^+$ axon patterns in adult *Pet1*cKO versus control spinal cords. We confirmed *Pet1* transcript loss in medullary 5-HT neurons by in situ hybridization (ISH) (*Figure 8—figure supplement 1A*). TdTomato$^+$ axon patterns from cervical to lumbar levels of the spinal cord were similar in *Pet1*cKO and control mice (*Figure 8A*). Quantitation of total TdTomato$^+$ axons in gray and white matter revealed no significant difference between *Pet1*cKO and control mice (*Figure 8B*). These findings indicate that Pet1, and consequently the hypothetical Lmx1b→Pet1 cascade, is not required for the formation of descending 5-HT axon pathways (*Figure 8—figure supplement 1E*).

We next investigated the *Pet1*cKO forebrain and found in striking contrast to the spinal cord that few if any TdTomato$^+$ axons were present in the olfactory bulb, cortex, amygdala, hippocampus, striatum and many regions of the hypothalamus in *Pet1*cKO forebrains (*Figure 8C–E*). Furthermore, the pattern of TdTomato$^+$ axons in the thalamus of *Pet1*cKO mice was severely disrupted with notable patches of clumped fibers and other areas with few, if any, fibers (*Figure 8—figure supplement 1B*). Similar to *Lmx1b*cKO mice, TdTomato$^+$ axons were present in normal density and with proper ascending trajectory within the MFB of *Pet1*cKO mice (*Figure 8—figure supplement 1C*). These findings clearly demonstrate a requirement for Pet1 in forebrain 5-HT pathway formation and thus distinct transcription factor requirements in the generation of ascending and descending 5-HT axon projection pathways.

Although *Pet1* conditional targeting largely phenocopies the forebrain 5-HT axon defects found in *Lmx1b*cKO mice, we noticed distinctly different patterns of TdTomato$^+$ arbors in the *Pet1*cKO vs. *Lmx1b*cKO thalamus (*Figure 8—figure supplement 1B*). The *Pet1*cKO thalamus lacked TdTomato$^+$ axons in lateral regions while in other thalamic regions we found abnormal clumping of *Pet1*cKO TdTomato$^+$ axons (*Figure 8—figure supplement 1B*). Further, in contrast to the failure of *Lmx1b*cKO axons to demarcate the PVT, *Pet1*cKO TdTomato$^+$ axons were properly patterned in this

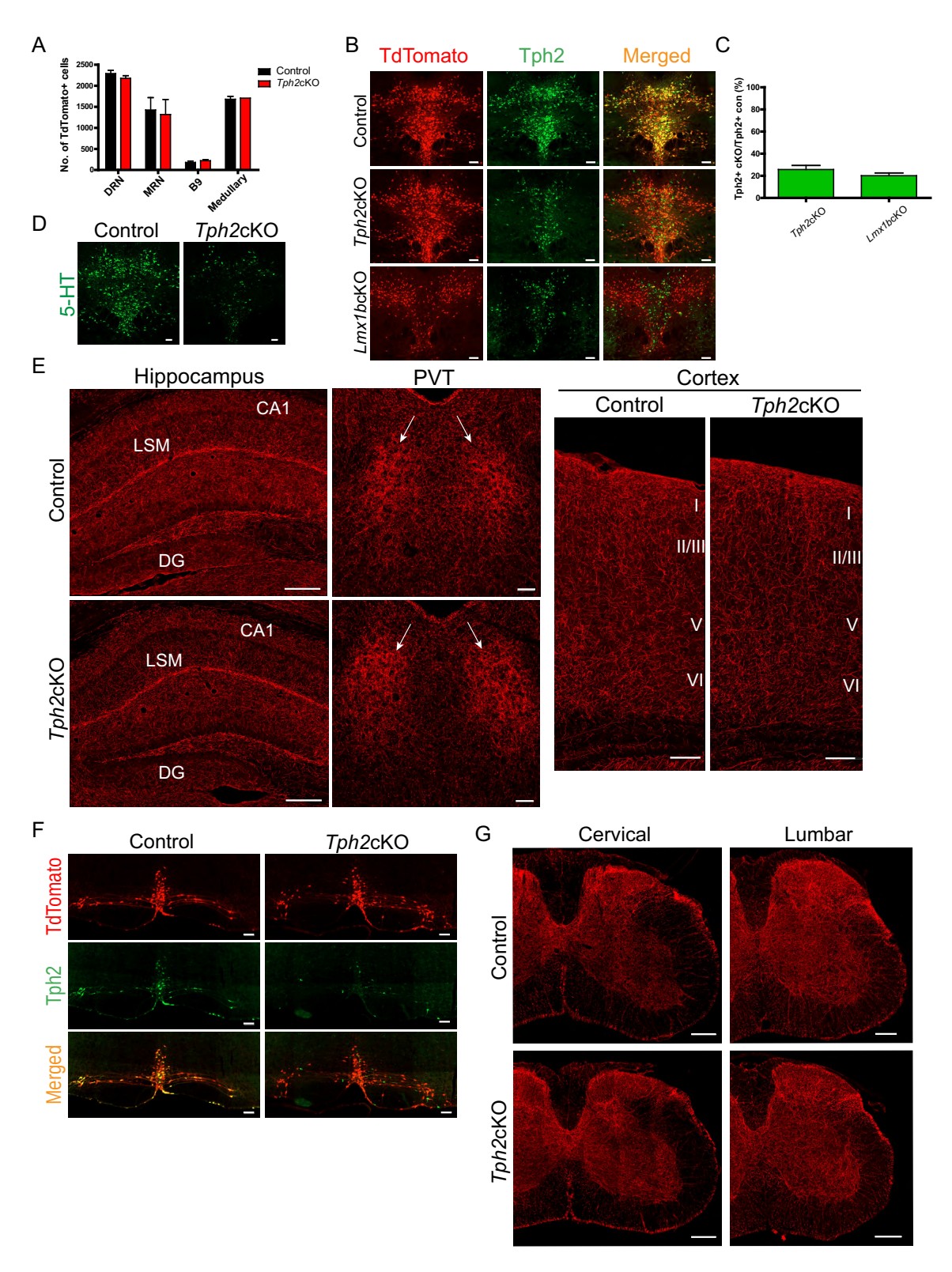

**Figure 6.** Specific targeting of 5-HT synthesis does not alter 5-HT arborization patterns. (**A**) Counts of TdTomato+ cells in each raphe nucleus of *Tph2*cKO mice did not differ from controls (n = 2 mice/genotype). Data are represented as mean ± SEM. (**B**) Comparable Tph2 knock-down in *Tph2*cKO and *Lmx1bc*KO mice. Scale bars, 100 μm. (**C**) Cell counts of residual Tph2+ neurons in *Tph2*cKO and *Lmx1bc*KO mice expressed as a percentage (n = 2 mice/genotype). Data are represented as mean ± SEM. (**D**) Immunolabeling shows 5-HT was severely reduced in *Tph2*cKO mice.
*Figure 6 continued on next page*

Figure 6 continued

Scale bars, 100 μm. (E) Coronal forebrain sections showing no deficits of TdTomato⁺ axon densities in *Tph2*cKO hippocampus, PVT, and cortex (n = 3 mice/genotype). LSM, lacunosum moleculare; DG, dentate gyrus; CA1 of hippocampus. Scale bars, 100 μm (PVT, cortex); 200 μm (hippocampus). (F) Co-immunolabeling for Tph2 and TdTomato in medullary neurons. Tph2 expression was severely reduced in medullary neurons of *Tph2*cKO mice. Scale bars, 50 μm. (G) No deficits of TdTomato⁺ axons were present throughout the *Tph2*cKO spinal cord (n = 3 mice/genotype). Scale bars, 200 μm.
DOI: https://doi.org/10.7554/eLife.48788.013

region (*Figure 8—figure supplement 1B*). This suggests Lmx1b may act independently of Pet1 in this specific highly discrete target field. Interestingly, early Pet1 deficiency did not result in reduced cell body size thus demonstrating that reduced cell body size did not contribute to the axon defects in *Lmx1b*cKO mice (*Figure 8—figure supplement 1D*). The robust similarities and yet distinct differences in TdTomato⁺ axon patterns in *Pet1*cKO compared to *Lmx1b*cKO forebrain support a model in which Lmx1b→Pet1 is the main regulatory program but that additional minor Lmx1b- or Pet1-orchestrated regulatory pathways operate in building ascending 5-HT axonal architectures (*Figure 8—figure supplement 1F*).

We investigated whether Lmx1b and Pet1 compensate for one another in the formation of 5-HT axons, by examining double-targeted mice (DKO: *Lmx1b*^fl/fl^; *Pet1*^fl/fl;Pet1-Cre;Ai9^) and controls (*Lmx1b*^+/+^; *Pet1*^+/+;Pet1-Cre;Ai9^). Analysis of DKO spinal cords revealed deficits in TdTomato⁺ axons patterns to similar levels as *Lmx1b*cKO spinal cord, further confirming Pet1 does not play a role in descending 5-HT axon development (*Figure 8—figure supplement 2A*). In the forebrain, we did not find a more extreme deficit in TdTomato⁺ axons in DKO mice compared to *Lmx1b*cKO mice. The DKO thalamus mirrored the *Lmx1b*cKO thalamus in the specific clumping of arbors and inability to properly pattern arbors in the PVT (*Figure 8—figure supplement 2B*).

To investigate whether the initial 1–2 days of Pet1 expression, not targeted by *Pet1-Cre* in *Pet1*cKO mice is required for primary ascending pathway formation, we analyzed *Pet1*^-/-;Pet1-YFP^ mice in which the *Pet1-YFP* transgene labels control and mutant Pet1⁺ cell bodies and their axons with YFP (*Hawthorne et al., 2010*). We found comparable initial 5-HT primary axon outgrowth in *Pet1*^-/-^ and control 5-HT neuron cell bodies at E13.5 (*Figure 8—figure supplement 2C*).

## Lmx1b acts through Pet1 to temporally control postnatal stage-specific gene expression and forebrain arborization

Stage-specific regulatory functions suggest potential stage-specific control of gene expression to fulfill successive steps in the morphological maturation of axon projection pathways. Thus, we next sought to determine whether the Lmx1b→Pet1 cascade temporally regulates expression of axon-related genes that are required at specific stages of 5-HT pathway formation. We found that Lmx1b continues to control Pet1 postnatally as *Pet1* transcript levels were decreased in P3 targeted *Lmx1b*icKO mice (*Figure 9A,B*). Next, we used RNA-sequencing to find common Lmx1b and Pet1 targets. Comparing rostral Lmx1b and Pet1 regulated genes, we found 82 regulated genes present in both datasets (*Figure 9C*). By intersecting rostral Lmx1b and Pet1 regulated genes with the set of Lmx1b-regulated axon-related genes (*Supplementary file 1*), we found 15 co-regulated genes (*Figure 9C*). Interestingly, one of these coregulated genes is *Pcdhac2*. Extensive studies of *Pcdhac2* have demonstrated its key intrinsic role in the formation and patterning 5-HT axon arbors (*Chen et al., 2017*; *Katori et al., 2009*; *Katori et al., 2017*). In particular, *Pcdhac2* deficient mice exhibit a lack of forebrain 5-HT arbors and severe clumping of remaining arbors, thus raising a functional link to Lmx1b and Pet1.

Since *Pcdhac2* is in the Pcdh alpha gene cluster and shares 3 (out of 4) exons with all other Pcdh alpha genes, we performed differential exon expression analysis with the Lmx1b and Pet1 RNA-seq datasets. By testing the only unique exon for *Pcdhac2*, we found that *Pcdhac2* was significantly regulated by Lmx1b and Pet1 at E17.5 (*Figure 9D*). In contrast, *Pcdhac2* expression was not significantly regulated by Pet1 at earlier stages (*Wyler et al., 2016*). Furthermore, our time-series RNA-seq analyses (*Wyler et al., 2016*) showed a significant and dramatic upregulation of *Pcdhac2* only at the onset of the arborization stage (*Figure 9E*). Thus, *Pcdhac2* expression is precisely controlled at the stage in which it is required for morphological maturation of 5-HT neurons.

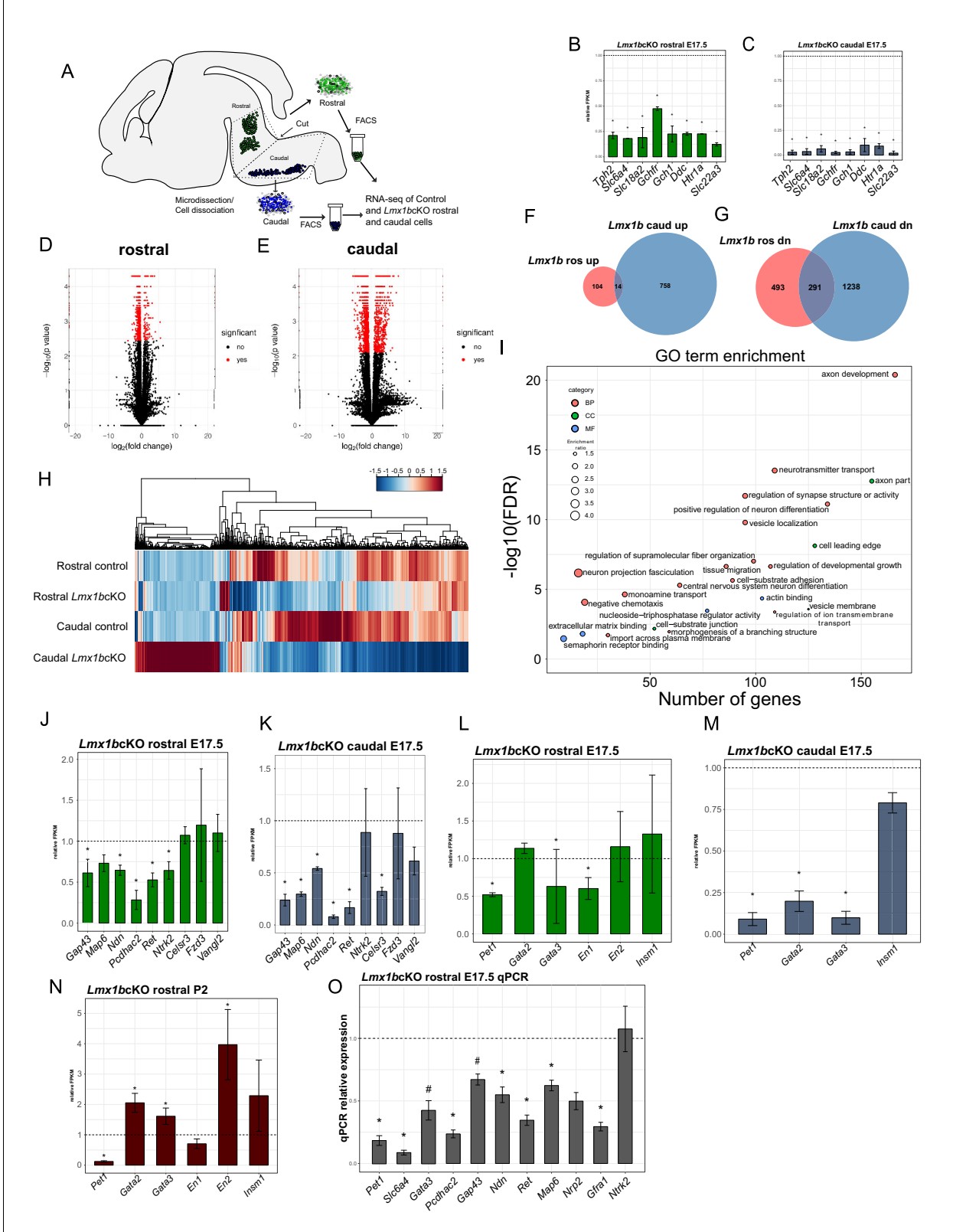

**Figure 7.** Ascending and descending Lmx1b regulated transcriptomes. (**A**) Schematic for dissection of E17.5 brain to isolate rostral and caudal 5-HT neurons genetically labeled by *Pet1-EYFP*. After dissection, EYFP$^+$ neurons were flow-sorted separately and prepared for RNA-sequencing (n = 3, controls; n = 2 *Lmx1b*cKO embryos). (**B**) Relative expression level of 5-HT pathway genes in rostral control versus *Lmx1b*cKO 5-HT neurons. Control gene expression levels were normalized to one. * indicates FDR ≤ 0.05. Data are represented as mean ± SEM. (**C**) Relative expression level of 5-HT

*Figure 7 continued on next page*

*Figure 7 continued*

pathway genes in caudal control versus *Lmx1bc*KO 5-HT neurons. Control gene expression levels were normalized to one. * indicates FDR ≤ 0.05. Data are represented as mean ± SEM. (D) Volcano plot for rostral control versus *Lmx1bc*KO differential expression. Significantly altered genes are in red with ≥log2(1.5X) and FDR ≤ 0.05. (E) Volcano plot for caudal control versus *Lmx1bc*KO differential expression. Significantly altered genes are in red with ≥log2(1.5X) and FDR ≤ 0.05. (F) Venn diagram of genes upregulated in rostral and caudal *Lmx1bc*KO 5-HT neurons. (G) Venn diagram of genes downregulated in rostral and caudal *Lmx1bc*KO 5-HT neurons. (H) Heatmap of differentially-expressed genes in rostral and caudal *Lmx1bc*KO 5-HT neurons. (I) GO term enrichment of Lmx1b regulated genes. BP = biological process, CC = cellular component, MF = molecular function. GO terms were enriched with FDR ≤ 0.05. (J) Relative expression (FPKMs) of known 5-HT neuron axon-related genes in rostral *Lmx1bc*KO 5-HT neurons. Data are represented as mean ± SEM. (K) Relative expression (FPKMs) of known 5-HT neuron axon-related genes in caudal *Lmx1bc*KO 5-HT neurons. Data are represented as mean ± SEM. (L) Relative expression (FPKMs) of 5-HT GRN transcription factors in rostral *Lmx1bc*KO 5-HT neurons. Data are represented as mean ± SEM. (M) Relative expression (FPKMs) of 5-HT GRN transcription factors in caudal *Lmx1bc*KO 5-HT neurons. Data are represented as mean ± SEM. (N) Relative expression (FPKMs) of 5-HT GRN transcription factors in flow sorted TdTomato$^+$ rostral *Lmx1bc*KO 5-HT neurons at postnatal day 2 (n = 3, controls; n = 4, *Lmx1bc*KO mice). Data are represented as mean ± SEM. (O) RT-qPCR verification of 5-HT GRN transcription factors and known axon-related genes from flow sorted rostral YFP+ *Lmx1bc*KO 5-HT neurons relative to control levels (n = 4 mice/ genotype). * indicates p-value≤0.05, # indicates p<0.1, t-test with Welch's correction. Data are represented as mean ± SEM.

DOI: https://doi.org/10.7554/eLife.48788.014

To determine whether Lmx1b was required to control the postnatal upregulation of *Pcdhac2*, we treated *Lmx1bic*KO mice with tamoxifen at postnatal day three and analyzed *Pcdhac2* expression by ISH at P14, when arborization is profusely developing. Indeed, we found substantially reduced expression of *Pcdhac2* in P3 *Lmx1bic*KO mice (*Figure 9F*). To further probe whether the Lmx1b→Pet1 regulatory cascade is required for upregulation of *Pcdhac2* during arborization, we generated *Pet1ic*KO (*Pet1*^fl/fl^; *Tph2-CreER*; Ai9) mice. We first verified that *Pet1* was effectively targeted in P3 *Pet1ic*KO mice by RT-qPCR of flow sorted TdTomato$^+$ cells as well as ISH for Pet1 (*Figure 9—figure supplement 1A,B*). We confirmed that similar numbers of TdTomato$^+$ neurons were activated in P3 *Pet1ic*KO mice (*Figure 9—figure supplement 1C,D*) and that long-range routes were filled compared to controls (*Figure 9—figure supplement 1E*). We found that Pet1 did not regulate Lmx1b at this stage (*Figure 9—figure supplement 1A*). Interestingly, we found that Pet1 was also required for the dramatic postnatal upregulation of *Pcdhac2* expression (*Figure 9G*). These findings show that the Lmx1b→Pet1 regulatory cascade acts during the arborization stage to temporally control *Pcdhac2* upregulation.

To determine if *Pcdhac2* is a direct target of Lmx1b→Pet1, we analyzed our previously published ChIP-seq datasets (*Wyler et al., 2016*) for mycPet1 binding sites within the *Pcdhac2* locus. Notably, we identified only two significant mycPet1 ChIP-seq peaks throughout the entire 250 kb Pcdha locus (*Figure 9H*). One peak was located at the 5' end of the unique *Pcdhac2* first exon. The second peak was located precisely within the third-intronic DNase I hypersensitivity site, HS7, which marks a well characterized transcriptional enhancer known to regulate midbrain expression of the Pcdha gene isoform (*Kehayova et al., 2011*; *Ribich et al., 2006*) (*Figure 9H*). Although the mycPet1 peak at the TSS of *Pcdhac2* does not contain a match to the known Pet1 binding motif, the second peak within HS7 does contain a significant match to the Pet1 motif (*Wei et al., 2010*) (*Figure 9I*).

To determine whether Lmx1b→Pet1 cascade acts postnatally to control terminal arborization we analyzed the forebrains of P3 targeted *Pet1ic*KO mice. Analyses performed at P49 revealed a severe deficit of 5-HT arbors in the hippocampus and cortex of P3 targeted *Pet1ic*KO mice, comparable to the deficit found in P3 targeted *Lmx1bic*KO mice (*Figures 5C*, *5D*; *Figures 9J*, *9K*). The timing of 5-HT terminal arborization varies widely in different regions of the early postnatal forebrain (*Lidov and Molliver, 1982*; *Maddaloni et al., 2017*). Thus, we next sought to determine whether the Lmx1b→Pet1 cascade continues to control arborization in forebrain regions that are late to develop their mature axon patterns. To investigate this, we administered a single dose of tamoxifen to *Pet1ic*KO pups at P5. We focused our subsequent analyses on the striatum, which is one of last regions of the forebrain to develop mature 5-HT arborization patterns (*Lidov and Molliver, 1982*). Interestingly, we found a notable deficit in the 5-HT arbors throughout the striatum (*Figure 9L*; *Figure 9—figure supplement 1F*). Together, our findings demonstrate that the Lmx1b→Pet1 cascade operates continually over an extended postnatal period to control stage-specific gene expression and to generate both early and late 5-HT terminal arbors patterns in different forebrain regions.

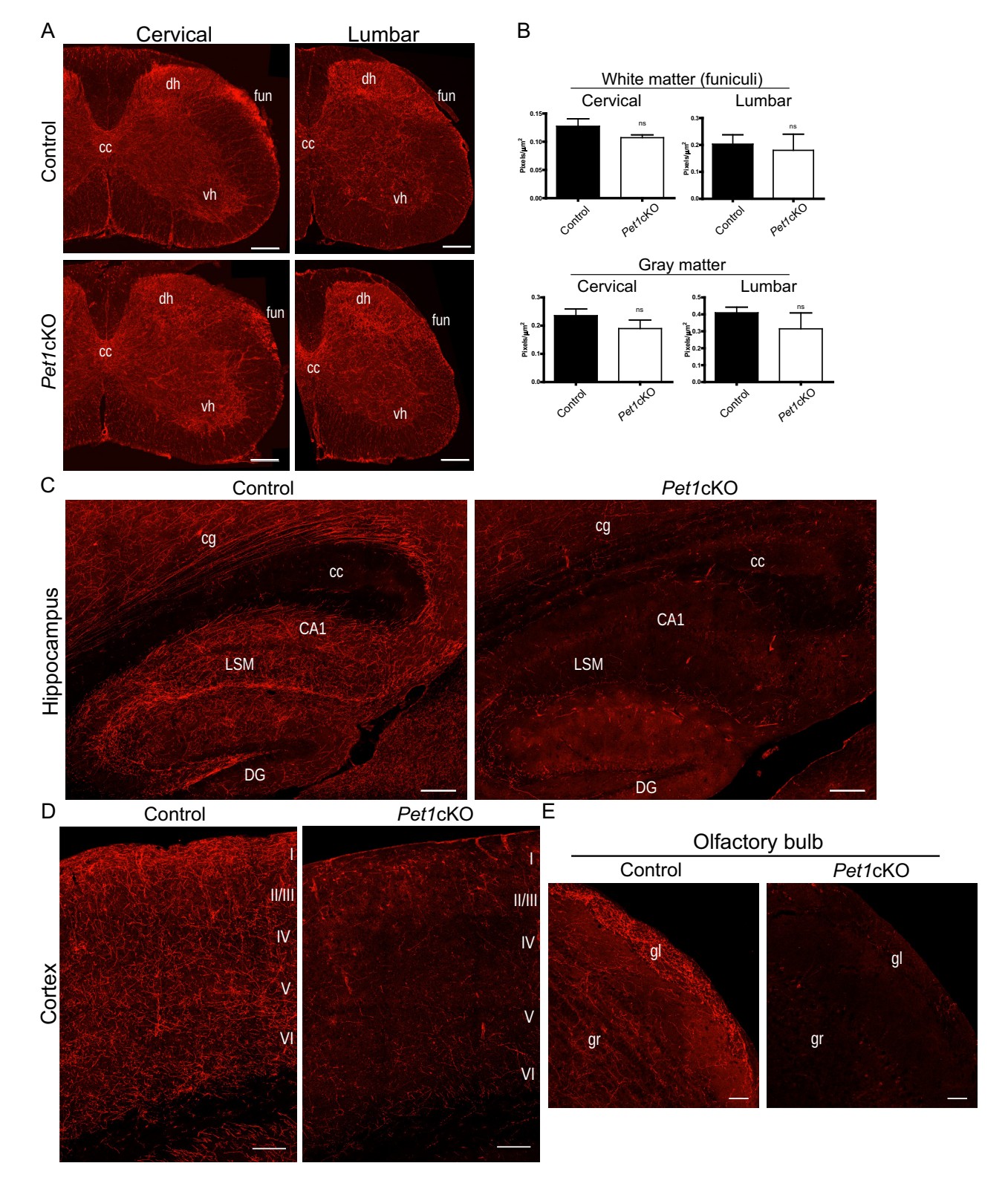

**Figure 8.** Distinct transcription factor requirements in the formation of ascending and descending 5-HT projection pathways. (**A**) TdTomato[+] axon innervation in *Pet1*cKO vs control spinal cords in 3 month old mice. Coronal semi-section views of cervical and lumbar levels. Scale bars, 200 μm. cc, *central canal*; dh, *dorsal horn*; vh, *ventral horn*; fun, *funiculi*. (**B**) Quantification of TdTomato[+] axons (pixels/μm²) in cervical and lumbar spinal cords (n = 3, controls; n = 3, *Pet1*cKO animals; Two-way ANOVA; white matter: cervical p=0.1372; lumbar p=0.6764; gray matter: cervical p=0.4440; lumbar

*Figure 8 continued on next page*

*Figure 8 continued*

p=0.1995). Data are represented as mean ± SEM. (**C**) Decreased TdTomato[+] arbors detected in *Pet1*cKO hippocampus compared to controls at 3 months of age. Scale bars, 200 µm, sagittal view. (**D**) Decreased TdTomato[+] arbors detected in *Pet1*cKO cortex compared to controls at 3 months of age. Scale bars, 100 µm, coronal view. (**E**) Decreased TdTomato[+] arbors detected in *Pet1*cKO olfactory bulb compared to controls at 3 months of age. Scale bars, 50 µm, sagittal view. cg, *cingulum*; cc, *corpus callosum*; LSM, *lacunosum moleculare*; DG, *dentate gyrus*; CA1 of hippocampus; gr, *granule layer*; gl, *glomerular layer*.

DOI: https://doi.org/10.7554/eLife.48788.015

The following figure supplements are available for figure 8:

**Figure supplement 1.** *Pet1*cKO and *Lmx1b*cKO mice exhibit distinct axon defects in thalamus.

DOI: https://doi.org/10.7554/eLife.48788.016

**Figure supplement 2.** DKO and *Pet1*[-/-] analyses.

DOI: https://doi.org/10.7554/eLife.48788.017

## Discussion

The process of long-range axon pathway formation occurs in temporally defined stages over an extended period during which successive morphological events occur (*Fame et al., 2011*; *Lidov and Molliver, 1982*; *Shirasaki and Pfaff, 2002*). Many regulatory factors have been implicated in the intrinsic control (*Santiago and Bashaw, 2014*) of axon outgrowth, target selection, and terminal arborization (*Arlotta et al., 2005*; *Chen et al., 2005*; *Galazo et al., 2016*; *Livet et al., 2002*; *Srivatsa et al., 2015*). What is not understood are the intrinsic programs that operate temporally to progressively control the prolonged, multistage process of long-range axon projection pathway formation (*Paolino et al., 2018*). Our findings uncover a temporal regulatory strategy through which a continuously expressed transcription factor, Lmx1b, operates at successive stages to control progressive steps in the postmitotic morphological maturation of long-range highly diffuse axonal projection pathways. Thus, Lmx1b through its continuous expression not only controls the capacity for 5-HT synthesis and reuptake (*Zhao et al., 2006*), but also the formation of long range profusely arborized projection pathways that enable delivery of the transmitter throughout the CNS.

Our findings suggest Lmx1b acts to successively control 5-HT axon primary outgrowth, selective routing, and terminal arborization. We uncovered a delay in primary 5-HT axon outgrowth between E16.5 to E18.5 in *Lmx1b*cKO mice. However, initiation of primary axon outgrowth toward the forebrain or spinal cord was not disrupted in either *Lmx1b*cKO, *Pet1*cKO, or *Pet1*[-/-] mice. As 5-HT axonogenesis occurs concomitant with the onset of 5-HT synthesis in newborn 5-HT neurons (*Hawthorne et al., 2010*), perhaps upstream regulatory programs operating at the progenitor stage control initial axon outgrowth from newborn 5-HT neurons (*Briscoe et al., 1999*; *Jacob et al., 2007*; *Pattyn et al., 2004*).

The profound defect in the subsequent selective 5-HT axon routing through the cingulum, SCS, and fornix suggests a failure of intrinsic growth extension beyond selective routing choice points. In support of this, we never observed ectopic 5-HT axon trajectories. Yet, we leave open the possibility that axon guidance defects contribute to the failure of 5-HT axons to selectively extend through proper routes. 5-HT axons normally turn dorsally through the septal area at E18.5, which might constitute a critical choice point in building expansive axon 5-HT architectures. In contrast, the vast majority of mutant 5-HT axons fail to make the dorsal turn and then fail to continue into selective tracts. In support of a possible guidance defect, our RNA-seq studies indicated that Lmx1b controls expression of a large number of genes encoding guidance receptors, guidance receptor ligands, and adhesion molecules.

Transcription factors function within GRNs as elucidated in the specification and differentiation of various types of neurons (*Deneris and Gaspar, 2018*; *Guillemot, 2007*; *Shirasaki and Pfaff, 2002*). How transcription factors temporally interact within networks to progressively generate precise connectivity patterns is poorly understood (*Paolino et al., 2018*). A focus of our experiments was to determine whether Lmx1b functions in the context of the 5-HT GRN to control progressive development of 5-HT axonal pathways. We uncovered a temporal requirement for Lmx1b in maintaining Pet1 expression during the early postnatal stage of 5-HT pathway formation. This finding together with postnatal temporal targeting of Pet1 revealed the Lmx1b→Pet1 regulatory cascade acts stage specifically to control selective pathway routing and arborization of forebrain 5-HT axons. Despite

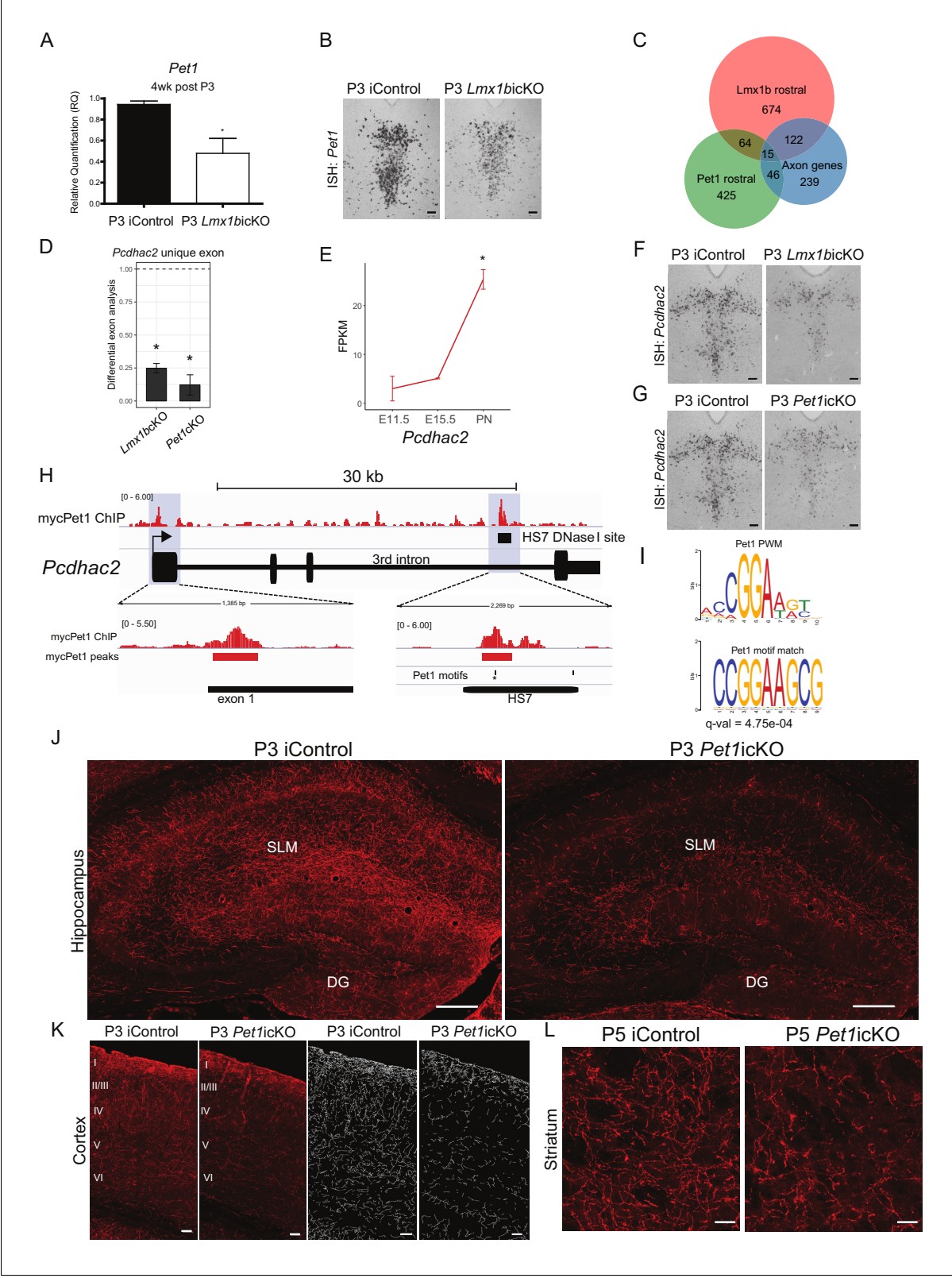

**Figure 9.** An ascending specific Lmx1b→Pet1 cascade controls stage specific 5-HT gene expression and postnatal terminal arborization. (**A**) RT-qPCR analysis of *Pet1* expression in flow sorted TdTomato⁺ neurons 4 weeks post P3 tamoxifen treatment (n = 3, iControl; n = 4, P3 *Lmx1b*icKO mice). Unpaired t-test with Welch's correction, *p<0.05. Data are represented as mean ± SEM. (**B**) *Pet1* in situ hybridization in P3 targeted *Lmx1b*icKO mice. Scale bars, 100 μm. (**C**) Venn diagram showing overlap of rostral Lmx1b and Pet1 regulated genes and the axon-related gene dataset. Lmx1b

*Figure 9 continued*

rostral: genes controlled by Lmx1b in rostral 5-HT neurons; Pet1 rostral: genes controlled by Pet1 in rostral 5-HT neurons; Axon genes: Lmx1b regulated rostral and caudal axon-related genes. (D) Relative expression of the unique *Pcdhac2* exon in rostral *Lmx1bc*KO and *Pet1c*KO 5-HT neurons. * indicates FDR ≤ 0.05. Data are represented as mean ± SEM. (E) Developmental expression profile of *Pcdhac2* in 5-HT neurons from E11.5 to early postnatal (PN). * indicates FDR ≤ 0.05. Data are represented as mean ± SEM. (F) *Pcdhac2* in situ hybridization at postnatal day 14 in P3 targeted *Lmx1bic*KO mice. Representative image from n = 3 mice/genotype. Scale bars, 100 μm. (G) *Pcdhac2* in situ hybridization at postnatal day 14 in P3 targeted *Pet1ic*KO mice. *ISH* experiments done in parallel, iControl section is the same section in (F) and (G) to compare icKOs at the same tissue level. Representative image from n = 3 *Pet1ic*KO mice. Scale bars, 100 μm. (H) Visualization of the *Pcdhac2* gene locus showing mycPet1 ChIP peaks at the TSS and 3$^{rd}$ intron (shaded) and matches to the known Pet1 motif. Zoomed regions of the significant mycPet1 binding sites are shown at the bottom. (I) The mycPet1 binding region located within the 3$^{rd}$ *Pcdhac2* intron DNAse I hypersensitivity site (HS7) contains a significant match to the known Pet1 position-weight matrix. TOMTOM q-value = 4.75×10$^{-04}$. (J) Decreased TdTomato$^+$ arbors detected in P3 targeted *Pet1ic*KO hippocampus compared to iControl mice. Scale bars, 200 μm. (K) Decreased TdTomato$^+$ arbors detected in all layers of P3 targeted *Pet1ic*KO cortex (Imaris tracing; *right panels*) compared to iControl mice. Scale bars, 50 μm. (L) Decreased TdTomato$^+$ arbors in striatum of P5 targeted *Pet1ic*KO mice. Scale bars, 20 μm. See also *Figure 9—figure supplement 1F*.

DOI: https://doi.org/10.7554/eLife.48788.018

The following figure supplement is available for figure 9:

**Figure supplement 1.** Lmx1b→Pet1 cascade acts postnatally to control 5-HT terminal arborization.

DOI: https://doi.org/10.7554/eLife.48788.019

the requirement for Pet1 in the acquisition of medullary 5-HT neuron-type identity (*Hendricks et al., 2003*; *Kiyasova et al., 2011*), it was surprising to find that Pet1 was not required for routing and arborization of descending 5-HT axons, thus revealing distinct transcription factor requirements in the generation of ascending and descending 5-HT axon pathways. Perhaps, Lmx1b operates with other continuously expressed serotonergic transcription factors in a regulatory cascade to control descending 5-HT axonal pathways. Gata3 is an interesting candidate as it functions more prominently in the medullary 5-HT neurons that generate the descending subsystem than 5-HT neurons that give rise to the ascending subsystem (*Pattyn et al., 2004*; *van Doorninck et al., 1999*).

The hundreds of axon related genes controlled by Lmx1b in the ascending and descending subsystems suggests that mis-expression of scores of functionally diverse effector genes accounts for the axon pathway defects we have reported. One gene of note is *Gap43*. Telencephalic commissures of *Gap43* null mice fail to form, including the hippocampal commissure and the corpus callosum (*Shen et al., 2002*). This may largely explain the 5-HT neuron axonal projection deficits reported in the *Gap43* null forebrain (*Donovan et al., 2002*). Although *Gap43* is expressed in 5-HT neurons, it is not yet known whether it plays an intrinsic role in 5-HT axon development (*Bendotti et al., 1991*). Our findings highlight the potential importance of an intrinsic Lmx1b→Pet1→Gap43→5-HT axon regulatory path given that *Gap43* expression is reduced in both *Lmx1bc*KO and *Pet1c*KO mice. It will be interesting to investigate this putative path in 5-HT conditionally targeted *Gap43* mice to determine whether it accounts for the selective routing defects present in *Lmx1bc*KO and *Pet1c*KO mice.

The stage specific regulatory roles we have uncovered suggests Lmx1b may temporally control certain genes to fulfill stage-specific events in the morphological maturation of 5-HT neurons. In support of this idea, our findings reveal that *Pcdhac2*, a key intrinsic effector of 5-HT terminal arbor growth and patterning (*Chen et al., 2017*; *Katori et al., 2009*; *Katori et al., 2017*), is dynamically regulated by the ascending Lmx1b→Pet1 cascade. Tamoxifen-inducible targeting of Lmx1b and Pet1 revealed a temporal requirement for Lmx1b and Pet1 in the postnatal upregulation of *Pcdhac2*. Notably, our ChIP-seq datasets revealed Pet1 occupancy at the *Pcdhac2*-specific promoter and HS7 suggesting Lmx1b→Pet1 directly controls upregulation of *Pcdhac2* during the postnatal arborization stage. Perhaps Pet1 occupancy at HS7 facilitates cohesin-mediated looping from HS7 to the *Pcdhac2* promoter thus accounting for Pet1 occupancy at this promoter in the absence of a high affinity Pet1 binding motif (*Guo et al., 2012*).

Deficient expression of a single stage-specifically expressed target gene such as *Pcdhac2* likely does not account for the severe and complex multi-stage defects in long-range 5-HT axon pathway formation reported here. However, our studies do serve to illustrate the concept that Lmx1b and Pet1 are critical terminal selectors of neuronal morphology and likely do so by dynamically regulating downstream genes that are required at specific stages for the progressive morphological maturation

of 5-HT neurons. Perhaps subsets of Lmx1b controlled genes act in different 5-HT neuron subtypes and at distinct stages to control the specific routes of 5-HT axons to diverse forebrain and spinal cord targets.

We speculate that our findings illustrate a possible general mechanism through which continuously expressed terminal selectors build long-range diffuse axon pathways. In addition to 5-HT neurons, noradrenergic, dopaminergic, and histaminergic neurons also generate expansive and highly arborized axonal architectures (*Björklund and Dunnett, 2007*; *Haas et al., 2008*; *Moore and Bloom, 1979*). Lmx1b plays a critical role in dopaminergic (DA) neuron development. However, in contrast to 5-HT neurons, Lmx1b is co-expressed in postmitotic DA neurons with its paralog, Lmx1a, up to about 2 months of age (*Laguna et al., 2015*). These two LIM HD factors play compensatory roles in mesDA neuron specification and differentiation (*Laguna et al., 2015*; *Yan et al., 2011*). Simultaneous DA targeting of Lmx1a and Lmx1b beginning at E14 resulted in diminished DA axon outgrowth to the dorsal striatum while DA axon projections to the ventral striatum and other DA axon targets throughout the forebrain remained intact (*Chabrat et al., 2017*). These findings suggest that temporal control of axon routing and arborization of the DA mesostriatal, mesolimbic and mesocortical projection pathways may be controlled by different continuously acting terminal selector transcription factors expressed in DA neurons.

In addition to their tremendous intrinsic capacity for long-range axonal growth and arborization during development, 5-HT neurons are noted for their rare ability to sprout and regenerate axons after injury (*Hawthorne et al., 2011*; *Jin et al., 2016*). Given Lmx1b's continuous expression into adulthood and crucial function in the formation of 5-HT projection pathways an intriguing line of investigation will be to determine whether Lmx1b transcriptionally powers 5-HT neuron's intrinsic potential for axon regrowth following injury in adult brain or spinal cord, which may suggest ways to harness that power for development of new repair strategies.

# Materials and methods

## Key resources table

| Reagent type (species) or resource | Designation | Source or reference | Identifiers | Additional information |
|---|---|---|---|---|
| Gene (*Mus musculus*) | *Lmx1b* | NA | MGI:1100513 | |
| Gene (*M. musculus*) | *Pet1* | NA | MGI:2449712 | |
| Gene (*M. musculus*) | *Tph2* | NA | MGI:2651811 | |
| Genetic reagent (*M. musculus*) | *Pet1-Cre* | PMID:16251278 | RRID:MGI:4837211 | Evan Deneris (Case Western Reserve University) |
| Genetic reagent (*M. musculus*) | *RosaTom* | Jackson Laboratory | Stock #: 007909; RRID:MGI:4436851 | Hongkui Zeng (Allen Institute for Brain Science) |
| Genetic reagent (*M. musculus*) | *Tph2-CreERT2* | Jackson Laboratory | Stock #: 016584; RRID:IMSR_JAX:016584 | Bernd Gloss (NIEHS) |
| Genetic reagent (*M. musculus*) | *Lmx1bflox* | PMID:17151281 | RRID:MGI:3810753 | Zhou-Feng Chen (Washington University) |
| Genetic reagent (*M. musculus*) | *Pet1flox* | PMID:20818386 | RRID:MGI:4837213 | Evan Deneris (Case Western Reserve University) |
| Genetic reagent (*M. musculus*) | *Tph2flox* | PMID:24972638 | RRID:IMSR_JAX:027590 | Zhou-Feng Chen (Washington University) |

*Continued on next page*

*Continued*

| Reagent type (species) or resource | Designation | Source or reference | Identifiers | Additional information |
|---|---|---|---|---|
| Genetic reagent (*M. musculus*) | *Pet1-/-* | PMID:12546819 | MGI:2449922 | Evan Deneris (Case Western Reserve University) |
| Genetic reagent (*M. musculus*) | *Pet1-YFP* | PMID:16251278 | n/a | Evan Deneris (Case Western Reserve University) |
| Antibody | anti-RFP (rabbit polyclonal) | Rockland | Rockland: p/n 600-401-379; RRID:AB_2209751 | (1:200) |
| Antibody | anti-GFP (rabbit polyclonal) | Invitrogen | Invitrogen: A6455; RRID:AB_221570 | (1:200) |
| Antibody | anti-5-HT (rabbit polyclonal) | Immunostar | Immunostar: 20080; RRID:AB_1624670 | (1:200) |
| Antibody | anti-Tph2 (rabbit polyclonal) | Millipore | Millipore: ABN60; RRID:AB_10806898 | (1:500) |
| Antibody | anti-Lmx1b (rabbit polyclonal) | *Suleiman et al., 2007*-gift | n/a | (1:200) |
| Antibody | anti-RFP (mouse monoclonal) | Abcam | Abcam: ab65856; RRID:AB_1141717 | (1:200) |
| Antibody | anti-RFP (mouse monoclonal) | Rockland | Rockland: p/n 200-301-379; RRID:AB_2611063 | (1:200) |
| Antibody | Alexa 488- or 594-secondaries | Invitrogen | | (1:500) |
| Genetic reagent (Virus) | rAAv2/Ef1a-DIO-hchR2 (H134R)-EYFP | UNC GTC Vector Core | | Lot# AV4378K |
| Sequence-based reagent | *Pcdhac2* in situ primer: F: 5' AGCCACCTCTAT CAGCTACCG 3' | this paper | | |
| Sequence-based reagent | *Pcdhac2* in situ primer: R: 5' AGAATTAACC CTCACTAAAGGGCTCAT TTTGAGAGCCAGCATCA 3' | this paper | | |
| Sequence-based reagent | Pet1 in situ primers: F:5'CCAGTGACCA ATCCCATCCTC3' | PMID:26843655 | | |
| Commercial assay or kit | Transcriptor First Strand cDNA Synthesis Kit | Roche | REF 4896866001 | |
| Commercial assay or kit | PerfeCTa PreAmp SuperMix | QuantoBio | Cat. No. 95146–040 | |
| Commercial assay or kit | PerfeCTa FastMix II ROX mastermix | QuantaBio | Cat. No. 95119–012 | |
| Commercial assay or kit | RNA Clean and ConcentratorTM-5 kit | Zymo Research | Catalog Nos. R1015 and R1016 | |
| Chemical compound, drug | Tamoxifen | Sigma-Aldrich | CAS Number: 10540-29-1 | 10 mg/mL stock in corn oil |
| Other | 35 μm filter | BD biosciences | Cat. No. 352235 | |
| Other | Fetal bovine serum (FBS) | Gibco | Cat. No. 16000044 | |
| Other | DNase I | Invitrogen | Cat. No. 18047019 | |
| Other | Trizol LS | Invitrogen | Cat. No. 10296010 | |

*Continued on next page*

*Continued*

| Reagent type (species) or resource | Designation | Source or reference | Identifiers | Additional information |
|---|---|---|---|---|
| Other | TrypLE | Invitrogen | Cat. No. 12605010 | |
| Other | Leibovitz's L15 medium | Invitrogen | Cat. No. 11415114 | |
| Commercial assay or kit | Quantifluor RNA system | Promega | Cat. No. E3310 | |
| Commercial assay or kit | Nugen TRIO RNA-seq | Nugen | Cat. No. 0507–08 | |
| Commercial assay or kit | Zymo RNA clean and concentrator | Zymo Research | Cat. No. R1013 | |
| Other | FACS Aria I | Becton Dickinson | | |
| Other | iCyt cell sorter | Sony | | |
| Other | Fragment analyzer | Advanced Analytics | | |
| Other | Nextseq 550 | Illumina | RRID:SCR_016381 | |
| Software, algorithm | FASTQC | https://www.bioinformatics.babraham.ac.uk/projects/fastqc/ | RRID:SCR_014583 | |
| Software, algorithm | Trimmomatic | *Bolger et al., 2014* | RRID:SCR_011848 | |
| Software, algorithm | Cufflinks (v2.2.2) | *Trapnell et al., 2010* | RRID:SCR_014597 | |
| Software, algorithm | R (v. 3.5) | http://www.r-project.org | RRID:SCR_001905 | |
| Software, algorithm | Webgestalt | http://www.webgestalt.org | RRID:SCR_006786 | |
| Software, algorithm | Biovenn | http://www.biovenn.nl | | |
| Software, algorithm | DEXSeq | *Anders et al., 2012* | RRID:SCR_012823 | |
| Software, algorithm | Tophat2 (v2.1.1) | *Kim et al., 2013* | RRID:SCR_013035 | |
| Other | *Mus musculus* genome | Ensembl, v. 96 | RRID:SCR_002344 | |
| Other | *Mus musculus* genome | UCSC, mm10 | RRID:SCR_005780 | |
| Software, algorithm | featureCounts | Rsubread | RRID:SCR_012919 | |
| Software, algorithm | ENCODE Transcription Factor ChIP-seq analysis pipeline | https://github.com/ENCODE-DCC/chip-seq-pipeline | RRID:SCR_015482 | |
| Software, algorithm | Burroughs-Wheeler aligner (BWA) | http://bio-bwa.sourceforge.net/ | RRID:SCR_010910 | |
| Software, algorithm | MACS2 | *Zhang et al., 2008* | RRID:SCR_013291 | |
| Software, algorithm | Imaris 7.4.2 | Bitplane, Zurich, Switzerland | | Surfaces or Filament Tracer tool |

## Animals

All procedures were approved by the Institutional Animal Care and Use Committees of Case Western Reserve University in accordance with the National Institutes of Health Guide for the *Care and Use of Laboratory Animals.* Experiments were performed on male and female mice using age-matched and sex-matched controls in triplicate unless otherwise noted. Adult mice between the ages 2.5 to 3.5 months were used unless otherwise noted. Early conditional knockout mice (designated *Lmx1bcKO, Pet1cKO,* and *Tph2cKO*) were generated by using the following genetic mouse lines: *Pet1-Cre* (original name *ePet-Cre*) (*Scott et al., 2005*), *Ai9 Rosa TdTomato* (*Rosa*$^{Tom}$; Jackson labs) and either *Lmx1b*$^{fl}$ (*Zhao et al., 2006*), *Pet1*$^{fl}$ (*Liu et al., 2010*), or *Tph2*$^{fl}$ (*Kim et al., 2014*). All early conditional knockout mice were compared to non-floxed controls (+/+; *Pet1-Cre;Ai9*). The following genetic lines were used to generate *Lmx1bicKO* or *Pet1icKO* mice for postnatal stage conditional knockout: *Tph2-CreER*$^{T2}$ (Jackson lab), *Ai9 Rosa TdTomato* (*Rosa*$^{Tom}$; Jackson lab) and either *Lmx1b*$^{fl}$ (*Zhao et al., 2006*) or *Pet1*$^{fl}$ (*Liu et al., 2010*). All postnatal stage

conditional knockout mice were compared to non-floxed controls (iControls: +/+;*Tph2-CreER*; Ai9). *Pet1*[-/-] mice (*Hendricks et al., 2003*) carrying the *Pet1-YFP* transgene (*Scott et al., 2005*) were used for embryo studies at E13.5. Tail or ear genomic DNA was used to determine genotypes of all animals. All mice were housed in ventilated cages on a 12 hr light/dark cycles with access to food and water with 2–5 mice per cage.

## Embryos and postnatal pups
All embryo experiments at each time point were performed in triplicate unless otherwise noted and compared to littermate controls (*Lmx1b*[fl/+] vs *Lmx1b*[fl/fl]; *Pet1*[fl/+] vs *Pet1*[fl/fl]; *Pet1*[+/-] vs *Pet1*[-/-]). Both male and female littermates were used for analysis. Embryonic day 0.5 (E0.5) was determined by presence of a vaginal plug. Postnatal day 0 (P0) was designated by date of birth.

## Histology and immunohistochemistry
Adult mice were anesthetized with Avertin (44 mM tribromoethanol, 2.5% tert-amyl alcohol, 0.02 ml/g body weight) and perfused for 2–3 min with cold PBS followed by 20 min cold 4% paraformaldehyde (PFA) in PBS. Brains and/or spinal cords were removed, post-fixed in 4%PFA for 2 hr and placed in 30% sucrose/PBS overnight (O/N) for cryoprotection. E12-E13 embryos were drop fixed in 4% PFA in PBS O/N followed by 30% sucrose incubation O/N. All embryos between age E15-E18 were transcardially perfused with 5 mL of 4% PFA in PBS, incubated in 4% PFA in PBS O/N, and incubated in 30% sucrose O/N. All tissue collected was frozen in Optimal Cutting Temperature (OCT) solution and sectioned on a cryostat at 25 µm. Tissue sections were mounted on SuperFrost Plus slides (Thermo Fisher Scientific) and vacuum dried. Sections were then permeabilized in 0.3% Triton 100X-PBS (PBS-T) for 15 min followed by antigen retrieval in Sodium Citrate buffer for 5 min in the microwave at low power. Sections were blocked with 10% NGS in PBS-T for 1 hr followed by incubation in primary antibody using a rabbit anti-RFP antibody (1:200; p/n 600-401-379, Rockland) at 4 ˚C O/N. Antigen retrieval step was not used for the following primary antibodies: rabbit anti-GFP (1:200; A6455, Invitrogen), rabbit anti-5-HT (1:200; Immunostar), rabbit anti-Tph2 (1:500; ABN60, Millipore), and rabbit anti-Lmx1b (1:200; *Suleiman et al., 2007*). For all co-stains with TdTomato a mouse anti-RFP (1:200; ab65856, Abcam) or a mouse anti-RFP (1:200; p/n 200-301-379, Rockland) primary antibody was used. Secondary antibodies were used at 1:500; goat anti-rabbit or mouse, Alexa Fluor 594 or 488 (Invitrogen).

## In situ hybridization
In situ hybridization was performed using a standard protocol using a digoxigen-11-UTP labeled antisense RNA probe to detect *Pet1* and *Pcdhac2* (Roche diagnostics) as described elsewhere (*Wyler et al., 2016*). *Pcdhac2* probe (566 bp) was generated using the following primers: F: 5' AGC-CACCTCTATCAGCTACCG 3' and R: 5' AGAATTAACCCTCACTAAAGGGCTCATTTTGAGAGC-CAGCATCA 3'. *Pet1* probe (513 bp) was generated using primers: F:5'CCAGTGACCAATCCCATCC TC3' and R: 5'AGAATTAACCCTCACTAAAGGGTTAATGGGGCTGAAAGGGATA3'.

## Cell counts
5-HT neurons were identified by Tph2 immunostaining. To calculate the *Pet1-Cre* efficiency, every 4[th] section was taken through the entire rostrocaudal extent of the DRN/MRN/B9 and medullary areas of control mice (n = 2 control mice) and RFP[+]/TPH2[+] and TPH2[+]/RFP[+] ratios were calculated. *Tph2-CreER* efficiency in postnatal tamoxifen injected pups was calculated from sections taken through the entire rostrocaudal extent of the DRN/MRN/B9 and expressed as a ratio of RFP[+]/TPH2[+] and TPH2[+]/RFP[+] cells (n = 4 iControl mice). A computer program that permits blinded manual cell counting was used (*Fox and Deneris, 2012*) to determine numbers of TdTomato[+] cells between control vs *Lmx1b*cKO, control vs *Tph2*cKO, and iControl vs *Lmx1b*icKO or *Pet1*icKO mice. Every 4[th] matched section was counted through the entire rostrocaudal extent of the DRN/MRN/B9 or medullary 5-HT system as described. To calculate the percentage of remaining Tph2[+] neurons in *Lmx1b*cKO vs *Tph2*cKO mice, Tph2[+] neurons were counted in every 4[th] section throughout the DRN/MRN/B9. Tph2[+] cell numbers in cKO animals was normalized to the number of Tph2[+] cells in control matched sections and expressed as a percentage for each genotype comparison.

## Tamoxifen injections

P1 pups (*Lmx1b*<sup>fl/fl;*Tph2*-CreER;Ai9</sup> and iControls; n = 4 mice/genotype) were injected 1X with 100 µg of tamoxifen (10 mg/mL stock in corn oil). P3 or P5 pups (P3: *Lmx1b*<sup>fl/fl;*Tph2*-CreER;Ai9</sup>, *Pet1*<sup>fl/fl;*Tph2*-CreER; Ai9</sup>, and iControls; n = 3 mice/genotype; P5: *Pet1*<sup>fl/fl;*Tph2*-CreER;Ai9</sup> and iControls, n = 2 mice/genotype) were injected 1X with 200 µg tamoxifen. All tamoxifen injections were performed subcutaneously at the back of the neck using a 30-gauge needle on a 1 mL syringe. Pups were allowed to sit for a few minutes before returning to mother. Injected mice were taken at either 31 or 49 days of age for analysis as indicated.

## Viral injections

Adult animals (*Lmx1b*<sup>fl/fl;*Pet1*-Cre;Ai9</sup> and *Lmx1b*<sup>+/+;*Pet1*-Cre;Ai9</sup>; n = 2 mice/genotype) were anesthetized with Isoflurane and stereotaxically injected unilaterally at two sites with 1.5 µl and 1 µl rAAv2/Ef1a-DIO-hchR2 (H134R)-EYFP (UNC GTC Vector Core Lot# AV4378K) at X = 0.6 mm, Y = −4.2 mm, Z = −3.2 mm and X = 0.6 mm, Y = −4.2 mm, Z = −4.2 mm relative to Bregma respectively. Animals were treated with a local anesthetic (bupivacaine HCL; Hospira) administered subcutaneously prior to surgery and with analgesic (carprofen 5 mg/Kg; Pfizer) for 3 days following. Holes were drilled through the skull to expose brain, following which a Hamilton syringe was slowly lowered to desired coordinates. Infusion rate was set to 0.1 µl/min with 10 min after each injection to allow diffusion of virus. Animals were returned to group housing following recovery and sacrificed 10 weeks after surgery.

## 5-HT neuron dissociation and flow cytometry

For RNAseq analyses rostral and caudal E17.5 YFP<sup>+</sup> 5-HT neurons were collected from *Lmx1b*<sup>fl/fl;*Pet1*-EYFP</sup> or *Pet1*<sup>fl/fl;*Pet1*-EYFP</sup> (controls) and *Lmx1b*<sup>fl/fl;*Pet1*-EYFP;*Pet1*-Cre</sup> or *Pet1*<sup>fl/fl;*Pet1*-EYFP;*Pet1*-Cre</sup> (*Lmx1b*cKO or *Pet1*cKO) mice using flow cytometry. To dissociate embryonic 5-HT neurons, hindbrains were initially dissected to separate rostral 5-HT neurons from caudal 5-HT neurons in cold PBS. Tissue was transferred to 1.5 mL ependorf tubes and centrifuged at 1500 rpm for 1 min at 4 °C. PBS was removed and 500 µL 1X TrypLE Express (Gibco) was added to the tissue and incubated at 37 °C for 15 min followed by addition of L-15 media (Gibco). Samples were centrifuged for 1 min at 1500 rpm and washed 3X with PBS. Tissue was then resuspended in 500 µL L15/0.1%BSA/DNase solution and slowly triturated 30X with fire-polished Pasteur pipettes of decreasing bore size until fully suspended. Samples were then filtered, and flow sorted on a FACS Aria I or Sony iCyt cell sorter.

Postnatal 5-HT neuron dissociation was performed in either P2 *Lmx1b*cKO or 4 week old *Lmx1b*icKO or *Pet1*icKO mice. Brains were sliced at 400 µm on a vibratome (Pelco easiSlicer) in continuously bubbling (95%$O_2$; 5%$CO_2$) aCSF solution (3.5 mM KCl, 126 mM NaCl, 20 mM NaHCO$_3$, 20 mM Dextrose, 1.25 mM NaH$_2$PO$_4$, 2 mM CaCl$_2$, 2 mM MgCl$_2$, 50 µm AP-V (Tocris), 20 µm DNQX (Sigma), and 100 nM TTX (Abcam)). Sections containing rostral 5-HT neurons were incubated in 1 mg/mL Protease from Streptomyces griseus (Sigma; P8811) in bubbling aCSF solution for 30min (P2 mice) or 45min (4-week-old mice) at room temperature. Slices were then incubated for 15min in bubbling aCSF alone at RT. TdTomato<sup>+</sup> neurons were microdissected from slices in cold aCSF/10%FBS solution. Samples were slowly triturated 30-100X with fire-polished Pasteur pipettes of decreasing bore size until fully suspended. Samples were then filtered, and flow sorted on a FACS ARIA-SORP sorter.

## RNA sequencing

Total RNA was isolated from flow-sorted neurons using Trizol LS (Invitrogen) and the RNA Clean and Concentrator-5 kit (Zymo Research). RNA concentration and quality was determined using Quantifluor RNA system (Promega) and Fragment analyzer (Advanced Analytics). Samples were converted to cDNA, depleted of rRNA transcripts, and amplified using the TRIO RNA-seq kit for mouse (Nugen Inc). Single-end sequencing was performed on a Nextseq 550 (Illumina) for 76 cycles. Read quality was assessed using FASTQC (https://www.bioinformatics.babraham.ac.uk/projects/fastqc/) and adapters were trimmed using Trimmomatic (*Bolger et al., 2014*). Filtered and trimmed reads were aligned to the mouse genome (mm10, UCSC) using Tophat2 (v2.1.1) (*Kim et al., 2013*).

## RNA sequencing analysis

Gene expression quantification and differential expression were analyzed using Cufflinks v2.2.2 (*Trapnell et al., 2010*). In all differential expression comparisons, a gene was called differentially expressed if fold-change was ≥1.5 and false discovery rate (FDR) was ≤5%. Hierarchical clustering of differentially expressed genes in rostral *Lmx1bc*KO and caudal *Lmx1bc*KO samples was performed in R (v. 3.5) using row-scaled values with Euclidean distance and complete linkage. The heatmap was plotted with the gplots library. Gene Ontology analysis was performed using Webgestalt (http://www.webgestalt.org), requiring five genes per category and FDR ≤ 5%. Venn diagrams were generated using Biovenn (http://www.biovenn.nl) (*Hulsen et al., 2008*; *Wang et al., 2017*). Differential exon expression analysis was performed to test for differences of the unique *Pcdhac2* first exon with DEXSeq (*Anders et al., 2012*). RNA-seq reads were mapped to the *Mus musculus* genome (Ensembl, v. 96) and reads falling within annotated exons were counted using featureCounts (Rsubread). DEXSeq default settings were used and significant exon usage differences were determined at FDR ≤ 5%.

## qPCR

RNA was isolated from YFP+ flow sorted cells in Trizol LS (Invitrogen) using chloroform extraction followed by the RNA Clean and Concentrator-5 kit (Zymo Research). cDNA was then synthesized using equal input RNA with the Transcriptor First Strand cDNA Synthesis Kit (Roche). cDNA was then amplified (8 PCR cycles) using PerfeCTa PreAmp SuperMix (QuantoBio) followed by ExoI digest to remove excess primers. RT-qPCR was performed using PerfeCTa FastMix II ROX mastermix (QuantaBio) with TaqMan probes (Thermofisher Scientific).

## Chromatin immunoprecipitation analysis

ChIP-seq data for mycPet1 was obtained from *Wyler et al. (2016)*. The data was re-analyzed using the ENCODE Transcription Factor ChIP-seq analysis pipeline (https://github.com/ENCODE-DCC/chip-seq-pipeline). Reads were mapped to the *Mus musculus* genome (UCSC mm10) using the Burroughs-Wheeler aligner (BWA) and peaks were called with MACS2 with FDR ≤ 1% (*Zhang et al., 2008*).

## Axon quantification

*Spinal cord:* Coronal sections of spinal cords were taken at 25 µm. Two sections were taken blindly from each cervical and lumbar spinal segment from each genotype (n = 3, control; n = 3, *Lmx1bc*KO mice) and (n = 3, control; n = 3, *Pet1*cKO). Whole gray area or whole white matter area was selected independently, and pixels were quantified using Zeiss 2.3 Image Analysis module. Pixel values were normalized to selected area ($µm^2$) for each section. Two-way ANOVA with Welch's correction analysis was performed. p values for each comparison are detailed in Fig. Legends.

*Forebrain*: Coronal sections of hippocampus, cortex, and axon tracts (SCS and cingulum) were taken and TdTomato$^+$ pixels were quantified using Zeiss 2.3 Image Analysis module in P3 targeted iControls (n = 3 non-littermate mice) and P3 *Lmx1bic*KO mice (n = 3 non-littermate mice). Pixel values were normalized to selected area ($µm^2$). Unpaired t-test with Welch's correction statistical analysis was performed. p values for each comparison are detailed in Fig. Legends.

## Cell volume analysis

Thick tissue sections were sliced at 150 µm and cleared using CUBIC clearing technique to increase the optical transparency of sections (*Susaki et al., 2015*). CUBIC reagent was prepared using urea (25 wt% final concentration), Quadrol (25 wt% final concentration), Triton X-100 (15 wt% final concentration) and $dH_2O$. Tissue sections were incubated in CUBIC reagent at 4°C overnight. To avoid the distortion of tissue by compression of the glass cover slip, Blu-Tack reusable adhesive was used to create a reservoir with a 0.3–0.5 mm depth. The transferred tissue sections were then sealed in CUBIC reagent using a glass cover slip for z-stack confocal imaging of the endogenous TdTomato fluorescence of cell bodies. The 3D neuron images were reconstructed and measured using the Surfaces or Filament Tracer tool in Imaris 7.4.2 (Bitplane, Zurich, Switzerland). Unpaired t-test with Welch's correction statistical analysis was performed. Sample size and p values for each comparison are detailed in Fig. Legends.

## Image acquisition and processing

Immuno-stained slides were imaged on an LSM800 confocal microscope (Carl Zeiss). All embryonic forebrain images were captured using a BZ-X700 fluorescence microscope (Keyence) or Olympus Optical BX51 microscope. Global brightness and contrast were edited across whole images equally between genotypes. Whole sagittal sections (*Figure 1A,B*) were acquired using confocal 10X objective and both control and *Lmx1b*cKO sections were processed in ImageJ and subjected to equal background subtraction followed by Lookup Table-Fire to enhance visualization of TdTomato$^+$ axon intensities in the forebrain. Axon tracing in *Figure 5D* and *Figure 9K* was performed using the *Filament Tracer* module in Imaris 7.4.2 (Bitplane, Zurich, Switzerland).

## Accession codes

All data generated in this study are deposited in NCBI GEO under accession code GSE130514.

## Acknowledgements

We thank Lynn Landmesser for key insights and suggestions throughout the course of this study. We thank Polyxeni Philippidou for her many helpful comments on the manuscript. We thank Wen-Cheng Xiong for the use of the BZ-X700 fluorescence microscope. We thank Randy Johnson for Lmx1b floxed mice. We thank Ralph Witzgall for the Lmx1b antibody. We thank Steven Wyler for the pup picture. This research was supported by the Genomics Core Facility and the Flow cytometry core at the CWRU School of Medicine. This research was supported by NIH grants P50 MH096972 and RO1 MH062723 to ESD.

## Additional information

### Funding

| Funder | Grant reference number | Author |
|---|---|---|
| National Institute of Mental Health | P50 MH096972 | Evan S Deneris |
| National Institute of Mental Health | RO1 MH062723 | Evan S Deneris |

The funders had no role in study design, data collection and interpretation, or the decision to submit the work for publication.

### Author contributions

Lauren J Donovan, Conceptualization, Validation, Investigation, Interpretation of data; William C Spencer, Conceptualization, Data curation, Formal analysis, Validation, Investigation, Writing— original draft, Writing—review and editing, Interpretation of data; Meagan M Kitt, Investigation, Performed stereotaxic injection of rAAV2/Ef1a-DIO-hchR2-EYFP and in situ hybridization; Brent A Eastman, Investigation, Performed histochemical analysis and imaging of DKO mice; Katherine J Lobur, Investigation, Methodology; Kexin Jiao, Investigation, Methodology, Performed cell body morphology analyses and measurements, Performed Imaris tracing and imaging; Jerry Silver, Writing—review and editing, Interpretation of data; Evan S Deneris, Conceptualization, Supervision, Funding acquisition, Writing—original draft, Writing—review and editing, Interpretation of data

### Author ORCIDs

Lauren J Donovan (iD) https://orcid.org/0000-0002-5622-7402
William C Spencer (iD) https://orcid.org/0000-0002-9700-8011
Evan S Deneris (iD) https://orcid.org/0000-0003-4211-9934

### Ethics

Animal experimentation: All animal procedures used in this study were in strict accordance with the Guide for the Care and Use of Laboratory Animals of the National Institutes of Health. The protocol

was approved by the Case Western Reserve University School of Medicine Institutional Animal Care and Use Committee (Animal Welfare Assurance Number A3145-01, protocol #: 2014-0044).

### Decision letter and Author response
Decision letter https://doi.org/10.7554/eLife.48788.027
Author response https://doi.org/10.7554/eLife.48788.028

## Additional files

### Supplementary files
• Supplementary file 1. Legend Lmx1b-regulated axon-related genes in rostral and caudal 5-HT neurons at E17.5. Axon genes that are also regulated by Pet1 in rostral 5-HT neurons at E17.5 are in bold.
DOI: https://doi.org/10.7554/eLife.48788.020
• Transparent reporting form
DOI: https://doi.org/10.7554/eLife.48788.021

### Data availability
Raw ChIP-seq data GEO accession: GSE74315. RNA-seq data generated in this study and ChIP-seq analysis are deposited in NCBI GEO under accession code GSE130514.

The following dataset was generated:

| Author(s) | Year | Dataset title | Dataset URL | Database and Identifier |
|---|---|---|---|---|
| Donovan LJ, Spencer WC, Kitt MM, Eastman BA, Lobur KJ, Jiao K, Silver J, Deneris ES | 2019 | Lmx1b is required at multiple stages to build expansive serotonergic axon architectures | http://www.ncbi.nlm.nih.gov/geo/query/acc.cgi?acc=GSE130514 | NCBI Gene Expression Omnibus, GSE130514 |

The following previously published dataset was used:

| Author(s) | Year | Dataset title | Dataset URL | Database and Identifier |
|---|---|---|---|---|
| Wyler SC, Spencer WC, Green NH, Rood BD, Crawford L, Craige C, Gresch P, McMahon DG, Beck SG, Deneris ES | 2016 | Pet-1 Switches Transcriptional Targets Postnatally to Regulate Maturation of Serotonin Neuron Excitability. | http://www.ncbi.nlm.nih.gov/geo/query/acc.cgi?acc=GSE74315 | NCBI Gene Expression Omnibus, GSE74315 |

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
