## [Decision Letter]

Thank you for submitting your article "Lmx1b is required at multiple stages to build expansive serotonergic axon architectures" for consideration by *eLife*. Your article has been reviewed by three peer reviewers, one of whom is a member of our Board of Reviewing Editors, and the evaluation has been overseen by Eve Marder as the Senior Editor. The following individuals involved in review of your submission have agreed to reveal their identity: Patricia Gaspar (Reviewer #2).

The reviewers have discussed the reviews with one another and the Reviewing Editor has drafted this decision to help you prepare a revised submission.

Summary:

The transcription factor Lmx1b is known to be required for the expression of genes that underlie the serotonergic cell fate. However, this terminal fate selector transcription factor stays on even after the transmitter identify of these neurons is determined, leading the authors to ask whether it might also have other functions in the later stages of development of these cells, specifically their axonal arborization. To address this question, the authors use a conditional Lmx1b allele combined with different Cre strategies and the Ai9 reporter line to visualize and FACS isolate Lmx1b deficient neurons. In a nice series of experiments they show that not only embryonic (E12.5) but also postnatal (P1, P3) cell-type specific knockout of Lmx1b is sufficient to disrupt specific aspects of serotonergic axonal arborization. The authors then FACS sort cKO neurons for RNAseq and find a program of gene expression enriched for axonal growth categories that they believe contributes to Lmx1b-dependent axon arborization. They compare list against *Pet1* cKO neurons and also a ChIP for myc-tagged Pet1 from an earlier study. They focus in a bit on one potential gene target that has previously been shown to affect axonal arborization, Pcdhac2. They validate the regulation of its expression at the time of 5HT neuron arborization and its dysregulation in the cKO neurons.

This is an interesting and very solid piece of work combining detailed anatomical descriptions in an impressive number of mouse mutants with a thorough genomic analysis that opens up the molecular mechanisms involved in the observed phenotypes. It makes the point that a given transcription factor can have very different roles and control different gene sets during cell specification and the maturation of the full axon arbors, which is an important part of a neuron's identity. Additionally, the paper provides a very useful resource on gene candidates that are potentially involved in serotonin neuron maturation during the early postnatal period. This is important considering the role of serotonin dysfunction in vulnerability to psychiatric disorders, and understanding what molecular mechanisms underlie the critical developmental period of this system.

Essential revisions:

1) The story would be more complete with some further mechanistic studies, though the reviewers agreed these were not required for publication of this story. Nonetheless we note here that reviewers suggested it would be interesting to know whether transgenic expression of Pcdhac2 was sufficient to rescue any of the mutant phenotypes, whether mutation of the Pet1 binding motif in the Pcdha enhancer would be sufficient to recapitulate the phenotypes, and whether more could be said about the ordering of the functions of Lhx1b versus Pet1 that could be derived from deeper analysis of the RNAseq data or a combination of the RNAseq and ChIP-seq data. If the authors have data or commentary on these points, it would add icing to the cake of a thorough study.

2) Figure 6 shows that 5-HT depletion Tph2-CKO has no effect on axon terminal distribution. One reviewer noted that this contradicts observations on a very similar mouse model by the Pasqualetti team. This reviewer also cited the recent paper of Pratelli et al., 2017, which showed that adult depletion of 5-HT had similar effects, which seemed a nice demonstration of the plasticity of this system. The possible reasons for this discrepancy are not discussed. Further, if possible, some quantification of the axon tract defects in the forebrain would add more strength to the description in this study.

3) Finally the reviewers raised a few points from the text that deserve revision or discussion:

– The statement that Lmx1 mutants have "axon primary outgrowth and selective routing defects " does not seem accurate. From the data provided, the outgrowth and routing of the serotonin raphe axons seem to be undisturbed during early embryonic stages and there are no clear abnormal trajectories later (at least in the data provided). This suggests that axon guidance and general axon targeting is overall normal, but that there is a stalling of growth along existing axon tracts- which starts to be visible at E16, there is also defective axon terminal branching with reduction in fiber density and aggregated axon terminals in some regions.

– To help integrate the new observations from this study into the larger literature on serotonin neuron development, the authors could provide a better perspective/discussion on how their phenotypes resemble or differ from the ones described in other mutants. For example, i) The phenotype of the Lmx1B seems to be a sort of mix of the *Gap43* KO and of the Pcadhc2 -KO, which makes sense given the fact that they are both down-regulated. ii) The phenotype of developing serotonin axons in the forebrain *Pet1* cKO (Figure 8) is very similar to that described by Kiyasova et al. in the full *Pet1*-KO.

– The normal morphology of the serotonin terminals in the PVT and intralaminar thalamic nuclei (Figure 8—figure supplement 1B) in the *Pet1* cKO could support the notion that these are not misguided 5-HT axons, but serotonin axons with less dependence on Pet for growth, in a way similar to what they suggest for the spinal cord projections.

– The transcriptional analysis reveals more widespread transcriptional changes in the caudal group than in the rostral raphe group, whereas the axonal alterations seem to predominate in the rostral group. Do the authors have a suggestion as to why this may be the case?

---

## [Author Response]

Essential revisions:1) The story would be more complete with some further mechanistic studies, though the reviewers agreed these were not required for publication of this story. Nonetheless we note here that reviewers suggested it would be interesting to know whether transgenic expression of Pcdhac2 was sufficient to rescue any of the mutant phenotypes, whether mutation of the Pet1 binding motif in the Pcdha enhancer would be sufficient to recapitulate the phenotypes, and whether more could be said about the ordering of the functions of Lhx1b versus Pet1 that could be derived from deeper analysis of the RNAseq data or a combination of the RNAseq and ChIP-seq data. If the authors have data or commentary on these points, it would add icing to the cake of a thorough study.

Several previous studies have clearly demonstrated the critical intrinsic role for *Pcdhac2* in 5-HT terminal arborization (Katori et al., 2009, Katori et al., 2017, and Chen et al., 2017). We utilized this thorough understanding of Pcdhac2 function to illustrate the idea that Lmx1b→Pet1 controls stage-specific expression of a gene that is required in a stage-specific manner for postnatal development of 5-HT axons.

Our RNA-seq studies suggest the 5-HT axon defects we report result from mis-expression of a large number of functionally diverse Lmx1b controlled growth and guidance effectors. The papers listed above and our analysis of *Pcdhac2* regulation identify this gene as a key effector of 5-HT axon development and a direct downstream target of Lmx1b→Pet1. Our primary interest going forward is to identify less obvious direct effectors of the Lmx1b-directed 5-HT axonal growth program. The challenge is to conceive an efficient experimental approach to accomplish this objective. We have considered a transgenic approach for rescue but assert its low throughput (quadruple compound allelic mice) makes it impractical. Perhaps rescue via viral expression in tamoxifen treated Lmx1biCKO or *Pet1*iCKO mice would offer greater throughput. We are continuing to explore various rescue approaches.

We agree an interesting experiment is to determine whether mutation of Pet1 binding motifs in *Pcdhac2* phenocopy 5-HT axon defects in Lmx1b or Pet1 targeted mice. We are currently discussing the optimal experimental design with which to edit Pet1 binding sites in the *Pcdhac2* promoter and/or intronic enhancer and generate mutant mice for future in vivo studies.

We have discussed at length how a deeper analysis of the existing RNA-seq and ChIP-seq data might reveal more about the ordering of Lmx1b and Pet1 functions and do not believe further conclusions can be made. It is possible to analyze the data for the Lmx1b homeodomain sequence motif as a preliminary step to determine whether Lmx1b could directly bind to either Lmx1b-regulated or Pet1-regulated genes. This type of analysis is useful in certain circumstances, but certainly would not show definitive binding of Lmx1b to the regulated gene. Ideally, we would perform ChIP-seq for Lmx1b. However, a ChIP-grade Lmx1b antibody is not available. We are working to generate a custom rabbit polyclonal antibody to Lmx1b and will eventually test it for the ability to immunoprecipitate Lmx1b bound chromatin. In future studies, we will further evaluate the direct regulation of Lmx1b and Pet1 target genes identified in this study to determine the ordering of Lmx1b and Pet1 functions in controlling 5-HT neuron axon development.

2) Figure 6 shows that 5-HT depletion Tph2-CKO has no effect on axon terminal distribution. One reviewer noted that this contradicts observations on a very similar mouse model by the Pasqualetti team. This reviewer also cited the recent paper of Pratelli et al., 2017, which showed that adult depletion of 5-HT had similar effects, which seemed a nice demonstration of the plasticity of this system. The possible reasons for this discrepancy are not discussed.

There is no contradiction although we understand how this might be construed as one. We fully agree the Pasqualetti group has clearly shown that total elimination of brain 5-HT levels leads to alter development of 5-HT axons (Migliarini et al., 2013 and Pratelli et al., 2017). However, it is essential to note a key difference between our study and theirs: the Pasqualetti group generated eGFP-knock in mice into the Tph2 locus, which resulted in a complete loss of Tph2 expression. In contrast, our Tph2 targeting with the ePet-Cre driver led to a quantitative loss of Tph2 that was comparable to the loss of Tph2 in Lmx1b CKO mice (see Figure 6B-D). Our objective was to determine whether the reduced, but not total, loss of Tph2 expression in Lmx1bCKO mice accounts for some of the 5-HT axon defects reported in these mice. Thus, we generated Tph2 targeted mice with the ePet-Cre driver as used in Lmx1bCKO mice to achieve a comparably reduced level of Tph2 expression. As reported in Figure 6, we saw no axon defects in our Tph2 targeted mice. Our conclusion is that the 5-HT axon defects observed in our Lmx1bCKO mice is not accounted for by the substantial but not total loss of Tph2 expression and consequent substantial but partial reduction of 5-HT levels in these mice. In light of the Pasqualetti findings, we conclude that Tph2 and 5-HT levels need to be reduced to an extent greater than we achieved in either our Lmx1bCKO or our Tph2CKO mice to see defects in 5-HT arborization. Perhaps, 5-HT has to be totally eliminated as achieved the by the Pasqualetti group to see effects on 5-HT axons. We have now included an assessment of our Tph2 finding relative to those of the Pasqualetti lab in the revised Results.

Further, if possible, some quantification of the axon tract defects in the forebrain would add more strength to the description in this study.

We now include quantification of Td-tomato+ axon tracts and arbors in terminal target fields of Lmx1biCKO mice. See revised Figure 4 and Figure 5—figure supplement 1. In new Figure 4, we added a new panel, Figure 4F, presenting quantitation of P1 targeted axon tracts. In order to accommodate the new quantification, we placed the original panel Figure 5B, showing coronal cingulum bundles and SCS tracts, into a new supplemental figure (Figure 5—figure supplement 1A) along with the new quantification of TdTomato+ axons in tracts as well as cortex and hippocampus. We have replaced Figure 5D and Figure 9K with improved representative images.

3) Finally the reviewers raised a few points from the text that deserve revision or discussion:– The statement that Lmx1 mutants have "axon primary outgrowth and selective routing defects " does not seem accurate. From the data provided, the outgrowth and routing of the serotonin raphe axons seem to be undisturbed during early embryonic stages and there are no clear abnormal trajectories later (at least in the data provided). This suggests that axon guidance and general axon targeting is overall normal, but that there is a stalling of growth along existing axon tracts- which starts to be visible at E16, there is also defective axon terminal branching with reduction in fiber density and aggregated axon terminals in some regions.

We maintain there is in fact a defect in primary stage 5-HT axon outgrowth as 5-HT axons exhibit delayed growth between E16.5 to E18.5. Further, we believe there is a profound defect at the subsequent selective 5-HT axon routing stage because multiple 5-HT routes including cingulum, SCS, and fornix are nearly completely lacking 5-HT axons. What is not entirely clear is whether the failure to selectively route is because of an inherent axonal growth defect or a defect in sensing guidance cues or perhaps even a failure in both processes. We agree our material gives the appearance of a failure of intrinsic growth potential because 5-HT axons initiate growth normally but then fail to extend at selective routing choice points. Moreover, we have never seen evidence in support of aberrant axon trajectories; all 5-HT axons seem to follow proper axon tracts up to a point. We wonder whether Lmx1b may be controlling a yet-to-be-defined intrinsic program that powers long distance 5-HT axonal growth capacity. However, we do not believe we can rule out axon guidance defects in the failure to selectively extend 5-HT axons through proper routes. 5-HT axons normally turn dorsally through the septal area at E18.5. However, the vast majority of mutant 5-HT axons fail to make the dorsal turn. We leave open the possibility that the failure to turn is a guidance defect at this particular area of the brain, which might constitute a critical choice point in building expansive axon 5-HT architectures. In support of a possible guidance defect, our RNA-seq studies indicated that Lmx1b controls expression of a large number of genes encoding guidance receptors and adhesion molecules. We discuss these points in the revised Discussion.

– To help integrate the new observations from this study into the larger literature on serotonin neuron development, the authors could provide a better perspective/discussion on how their phenotypes, resemble or differ from the ones described in other mutants. For example, i) The phenotype of the Lmx1B seems to be a sort of mix of the Gap43 KO and of the Pcadhc2 -KO, which makes sense given the fact that they are both down-regulated. ii) The phenotype of developing serotonin axons in the forebrain Pet1 cKO (Figure 8) is very similar to that described by Kiyasova et al. in the full Pet1-KO.

i) We agree with the reviewers that the 5-HT axon forebrain phenotype in *Gap43* null mice looks outwardly similar to that reported in our Lmx1bCKO and *Pet1*cKO mice. This raises the possibility that loss of *Gap43* in our cko mice accounts for part of the axons defects we report. However, it is not yet known whether *Gap43* plays an intrinsic role in 5-HT axon development. Moreover, Shen et al., 2002 reported that *Gap43* null mice have a devastating and widespread disruption of brain development: these mice are missing the hippocampal commissure (HC) and the corpus callosum (CC). These severe structural abnormalities could in fact be the primary cause of the failure of 5-HT axons to innervate cortical and hippocampal regions of *Gap43* null mice (Donovan et al., 2002) because a majority of growing 5-HT axons use the HC and CC to route to these terminal target regions. However, *Gap43* is expressed in 5-HT neurons, which raises the possibility that it also plays an intrinsic role in 5-HT axon development. Our findings highlight the potential importance of an intrinsic Lmx1b→Pet1→*Gap43*→5-HT axon regulatory path given that *Gap43* expression is reduced in both Lmx1bCKO and *Pet1*CKO mice. It will be interesting to investigate this putative path in 5-HT conditionally targeted *Gap43* mice using the ePet-Cre driver and yet to be generated mice carrying a *Gap43* conditional allele. We discuss this in the revised Discussion.

ii) We have discussed this comment at length. Pet1 null mice lack 5-HT immunopositive axon fibers in many forebrain terminal target fields, which seemingly parallels the lack of Td-Tomato+ axons in these regions of Lmx1bCKO and *Pet1*CKO mice. However, it is difficult to directly compare the two mouse models. Kiyasova et al. focused on the minor residual population of 5-HT immunopositive axons present in the forebrain of Pet1 null mice. In our present study, we used the Ai9 reporter locus to mark and assess both 5-HT immunopositive axons as well as axons that have lost their 5-HT marker content because of Pet1 conditional targeting. Using the *Pet1*cKO mouse we discovered that Pet1 mutant Td-tomato+ axons, regardless of whether or not they are still 5-HT immunopositive, are unable to extend to the cortex or hippocampus.

– The normal morphology of the serotonin terminals in the PVT and intralaminar thalamic nuclei (Figure 8—figure supplement 1B) in the Pet1 cKO could support the notion that these are not misguided 5-HT axons, but serotonin axons with less dependence on Pet for growth, in a way similar to what they suggest for the spinal cord projections.

Yes, we agree with the reviewer’s interpretation. We find it interesting that the PVT is innervated by 5-HT mutant axons in *Pet1*cKO animals. We do believe that these axons properly reached the PVT in absence of Pet1, similar to spinal projections. This suggests Lmx1b might be acting independently of Pet1 in this specific highly discrete target field. In contrast, we believe *Pet1*cKO 5-HT axons do not properly innervate some other intralaminar thalamic nuclei (ITN) because we found an increased density of mutant axons in discrete areas of the ITN compared to controls (Figure 8—figure supplement 1B, asterisks). We have revised the Results subsection “An ascending-specific axonal Lmx1b→Pet1 regulatory cascade” to clarify this point.

– The transcriptional analysis reveals more widespread transcriptional changes in the caudal group than in the rostral raphe group, whereas the axonal alterations seem to predominate in the rostral group. Do the authors have a suggestion as to why this may be the case?

It is not clear why there are more widespread transcriptional changes in the caudal 5-HT neuron group than the rostral 5-HT group. It is not due to differences in Cre activity since we obtained equivalent Cre recombination efficiencies in both rostral (Figure 1—figure supplement 1) and caudal groups (Figure 2—figure supplement 1). We suggest one possibility is that there is greater cell sub-type diversity in the rostral group, which also has nearly twice the number of cells. Therefore, with more diversity, differences in RNA levels could be averaged out in the rostral group versus a more homogeneous caudal group.